# Climate influences on flood probabilities across Europe

Eva Steirou[1], Lars Gerlitz[1], Heiko Apel[1], Xun Sun[2, 3], Bruno Merz[1, 4]

[1]Section Hydrology, GFZ German Research Center for Geosciences, Potsdam, 14473, Germany
[2]Key Laboratory of Geographic Information Science (Ministry of Education), East China Normal University, 200241, Shanghai, China
[3]Columbia Water Center, Earth Institute, Columbia University, New York, NY 10027, USA
[4]Institute of Environmental Science and Geography, University of Potsdam, Potsdam, 14476, Germany

*Correspondence to*: Eva Steirou (esteirou@gfz-potsdam.de), Xun Sun (xs2226@columbia.edu)

**Abstract.** The link between streamflow extremes and climatology has been widely studied during the last decades. However, a study investigating the effect of large-scale circulation variations on the distribution of seasonal discharge extremes at the European level is missing. Here we fit a climate-informed Generalized Extreme Value distribution (GEV) to about 600 streamflow records in Europe for each of the standard seasons, i.e. to winter, spring, summer and autumn maxima, and compare it with the classical GEV with parameters invariant in time. The study adopts a Bayesian framework and covers the period 1950 to 2016. Five indices with proven influence on the European climate are examined independently as covariates, namely the North Atlantic Oscillation (NAO), the East Atlantic pattern (EA), the East Atlantic / West Russian pattern (EA/WR), the Scandinavia pattern (SCA) and the Polar-Eurasian pattern (POL).

It is found that for a high percentage of stations the climate-informed model is preferred to the classical model. Particularly for NAO during winter, a strong influence on streamflow extremes is detected for large parts of Europe (preferred to the classical GEV for 46% of the stations). Climate-informed fits are characterized by spatial coherence and form patterns that resemble relations between the climate indices and seasonal precipitation, suggesting a prominent role of the considered circulation modes for flood generation. For certain regions, such as Northwest Scandinavia and the British Isles, yearly variations of the mean seasonal climate indices result in considerably different extreme value distributions and thus in highly different flood estimates for individual years that can also persist for longer time periods.

## 1.    Introduction

The understanding of extreme streamflow is a key issue for infrastructure design, flood risk management and (re-) insurance, and the estimation of flood probabilities has been in the focus of the scientific debate during recent decades. Traditionally, streamflow has been analyzed with regard to associated hydro-climatic processes acting at the catchment scale. During recent years many studies have additionally focused on the link between local streamflow and larger-scale climate mechanisms, extending beyond the catchment boundaries (Merz et al., 2014). An early example can be found in Hirschboeck (1988), who provides a detailed explanation of relationships between floods and synoptic patterns in the USA. Large-scale atmospheric patterns acting at global or continental scales have been shown to significantly influence flood magnitude and frequency at the local and regional scale. Regional in this context refers to the joint consideration of several gauges. For example, Kiem et al. (2003) stratified a regional flood index in Australia according to quantiles of the El Niño/Southern Oscillation (ENSO) index and showed that La Niña events are associated with a distinctly higher flood risk compared with El Niño events. Ward et al. (2014) found that peak discharges are strongly influenced by ENSO for a large fraction of catchments across the globe. Delgado et al. (2012) detected a dependence between the variance of the annual maximum flow at stations along the Mekong River and the intensity of the Western Pacific monsoon.

This perception of climate-influenced extremes has been incorporated in flood frequency analysis by including climatic variables as covariates of extreme value distribution parameters. It is therefore assumed that the probability density function (pdf) of streamflow is not constant in time but it is conditioned on external variables. This framework, usually called

nonstationary, can be particularly useful for hydro-climatic studies since the influence of the climatic phenomena on the distribution of the hydrological target variable, such as extreme streamflow, can be considered (Sun et al., 2014). This means that the whole distribution as well as certain parts of the target variable distribution, such as the tails, can be assessed including the influence of the large scale climate phenomenon, and used for flood risk management or reinsurance purposes. This conditional or nonstationary frequency analysis has been popularized in the field of hydrology and flood research during recent years. Different covariate types have been examined for their influence on flood extremes, e.g. time (e.g. Delgado et al., 2010; Sun et al., 2015), snow cover indices (Kwon et al., 2008), reservoir indices (López and Francés, 2013; Silva et al., 2017), population measures (Villarini et al., 2009) and large-scale atmospheric and oceanic fields and indices (Delgado et al., 2014; Renard and Lall, 2014). A review of nonstationary approaches for local frequency analyses is given by Khaliq et al. (2006), while some of their limitations are discussed by Koutsoyiannis and Montanari (2015) and Serinaldi and Kilsby (2015) and Serinaldi et al. (2018).

In this study, we focus on the European continent and the relation between streamflow extremes and the large-scale atmospheric circulation. The European climate is mainly influenced by pressure patterns acting at the broader region covering Europe and the northern Atlantic. In particular, five circulation modes have been shown to significantly modify the moisture fluxes into the European domain: the North Atlantic Oscillation (NAO), the East Atlantic (EA), the East Atlantic/Western Russia (EA/WR), the Scandinavia (SCA) and the Polar/Eurasia (POL) patterns (Bartolini et al., 2010; Casanueva et al., 2014; Rust et al., 2015; Steirou et al., 2017). These patterns represent the first five pressure modes north of 50°, derived by means of a rotated principle component analysis of monthly mean 500hPa geopotential height fields (Barnston and Livezey, 1987). The modes indicate the position and magnitude of large-scale atmospheric waves and thus control the strength and location of the northern hemispheric Jetstream. All modes are characterized by a particular pattern of large-scale winds and moisture fluxes and strongly affect near-surface climate conditions over vast parts of the northern hemisphere. Particularly NAO has been shown to significantly influence the European winter climate: its positive state has been linked to positive (negative) anomalies of moisture fluxes, cyclone passages and precipitation over northern (southern) Europe (Hurrell and Deser, 2009; Wibig, 1999). A seasonal shift of the NAO pressure centers and moisture fluxes towards north during summer has been detected (Hurrell and Deser, 2009). EA, often referred to as a southward shifted NAO, is characterized by distinctly defined geopotential height anomalies and an associated influence on westerly moisture fluxes and local climate conditions over Great Britain (Comas-Bru and McDermott, 2014; Moore and Renfrew, 2012). EA/WR features two centers of action over Central Europe and Central Russia. During its positive state, a planetary ridge is located over north-western Europe, and this reduces the advection of moist air masses (Krichak and Alpert, 2005). SCA is particularly active over northern Europe and triggers atmospheric blocking during its positive phase (Bueh and Nakamura, 2007). POL represents the strength of the pressure gradient between the polar regions and the mid-latitudes and thus controls the westerly circulation, particularly over northern Europe (Claud et al., 2007). Correlation maps, demonstrating links between these circulation modes and seasonal precipitation and temperature, are included in the Supplementary Material (Fig. S1-S4).

Apart from Northern Hemisphere modes, the El Niño-Southern Oscillation (ENSO) has been suggested to influence the European hydrology. Significant relations have been found with precipitation and different discharge indices (Guimarães Nobre et al., 2017; Mariotti et al., 2002; Steirou et al., 2017). However, in contrast to the above described circulation modes, ENSO does not shape the European climate and hydrology directly, but rather indirectly through the regulation of the phase of other large-scale modes, such as the EA (Iglesias et al., 2014). Other patterns acting at a smaller scale, such as the Mediterranean Oscillation (MO) and the Western Mediterranean Oscillation (WMO), have also been related with hydrological variables in Europe (Criado-Aldeanueva and Soto-Navarro, 2013; Dünkeloh and Jacobeit, 2003; Martin-Vide and Lopez-Bustins, 2006). However, such modes seem to have limited importance at the continental scale.

While the relation between European hydrology and large-scale circulation has attracted much attention and has been widely studied, only few studies have adopted a conditional flood frequency framework for the investigation of climate-flood

interactions. Villarini et al. (2012) conducted a frequency analysis of annual maximum and peak-over-threshold discharge in Austria with NAO as a covariate. López and Francés (2013) examined maximum annual flows in Spain conditioned on the principal components of four winter climate modes: NAO, AO, MO and WMO. Still, a comprehensive study on streamflow extremes at the European scale has not been conducted.

Thus, this study aims at a large-scale investigation of circulation-streamflow interactions for the entire European continent by adopting a flood frequency framework. We examine seasonal streamflow maxima from more than 600 gauges covering the entire European continent and particularly investigate the influence of the five major pressure modes that directly affect the European climate: NAO, EA, EA/WR, SCA and POL. In order to quantify the effect of important hydro-climatological processes for the streamflow regimes, we investigate contemporaneous relationships only, without considering any time lags. We identify regions with a consistent influence of each particular circulation index in order to explain the spatial coherence of flood frequency. The analysis is conducted at a seasonal scale in order to better account for the intra-annual variations of the circulation characteristics and the associated seasonal shift of climate-streamflow relationships. A Bayesian framework is adopted for the flood frequency analysis because of its advantages concerning the quantification and interpretation of uncertainty. Furthermore, prior information about hydrologic extremes exists in the literature and can be used for inference.

## 2.    Data and Methods

### 2.1    Streamflow data and circulation indices

The time period of our analysis is from 1950 to 2016, defined by the overlap between streamflow data and circulation indices. Daily streamflow data for the European continent were received from GRDC (Global Runoff Data Centre). From this dataset, gauges with record lengths of at least 50 years after 1950 and with a catchment area larger than 200 km$^2$ were selected. Small catchments are not considered, as they may be more prone to local phenomena, which could blur the large-scale atmospheric influence. In total, 649 stations covering North and Central Europe with the exception of Poland are considered. Due to the underrepresentation of Southern Europe, additional data from other sources satisfying the above mentioned criteria are included in the analysis. Five time series with monthly maximum discharges were obtained for Spain and one station with daily discharge was provided for Portugal. For details about these additional stations the reader is referred to Mediero et al. (2014, 2015). Finally, one record with daily streamflow data was provided for Pontelagoscuro in Italy (Domeneghetti, 2017, personal communication). For each station, the maximum value of mean daily streamflow is derived for the four standard boreal seasons: winter (DJF), spring, (MAM), summer (JJA) and autumn (SON). Seasons with more than 20% missing values are not considered. Overall 586 records in winter, 604 in spring, 599 in summer and 597 for the autumn season are utilized for the analysis.

Time series of monthly circulation indices for the period 1950-2016 were retrieved from the Climate Prediction Center (CPC) of the National Oceanic and Atmospheric Administration (NOAA), (http://www.cpc.ncep.noaa.gov/data/teledoc/telecontents.shtml). We make use of the five indices mentioned in the introduction, namely, the NAO, EA, EA/WR, SCA and POL patterns. Seasonal mean climate indices are used for the adjustment of the extreme value distribution, however, we also examine whether the results differ if monthly values (in accordance with the observed flood date) are considered as covariate. The time series of the seasonal indices, along with their running mean for a 10-year window, are shown in Fig. S5. Histograms showing the distribution of mean circulation indices for each season are provided in Fig. S6.

### 2.2    Flood frequency analysis – Competing models

The GEV with parameters invariant in time and with parameters conditioned on the climate indices are fitted to the seasonal maximum streamflow data. For the climate-informed models the condition of independent and identically distributed

observations of the classical GEV is relaxed to include parameters conditioned on time-varying covariates (Katz et al., 2002).

For the two types of models we use the terms "classical model" instead of stationary model and "climate-informed model" rather than "nonstationary model". It has been suggested that if covariates have a stochastic structure and no deterministic component, the resulting distribution is not truly nonstationary (Montanari and Koutsoyiannis, 2014; van Montfort and van Putten, 2002; Serinaldi and Kilsby, 2015). As our climate covariates have no distinguishable deterministic component (not shown), it is consequently not clear if they result in nonstationary models. Here each streamflow gauge is handled

independently and site-specific parameters are derived. Let $Y(t)$ denote a streamflow observation at time $t$ and $\mathbf{Y} = (Y(t_1), Y(t_2), \dots, Y(t_n))$ denote the vector of streamflow observations at a specific site. Then for the classical case the model is given as:

$$Y(t) \sim GEV(\boldsymbol{\theta}) \tag{1}$$

where $\boldsymbol{\theta}$ is the vector of length $m$ of (time-invariant) distribution parameters. The classical GEV comprises $m = 3$ parameters; a location parameter $\mu$, a scale parameter $\sigma$ and a shape parameter $\xi$.

In the Bayesian framework, the posterior pdf of the parameter vector is computed as follows, based on Bayes theorem:

$$f(\boldsymbol{\theta}|\mathbf{Y}) \propto f(\mathbf{Y}|\boldsymbol{\theta})f(\boldsymbol{\theta}) \tag{2}$$

where f($\boldsymbol{\theta}$) is the prior pdf of distribution parameters and f($\mathbf{Y}|\boldsymbol{\theta}$) is the likelihood function:

$$f(\mathbf{Y}|\boldsymbol{\theta}) = \prod_t f(Y(t)|\boldsymbol{\theta}) \tag{3}$$

For the climate-informed distribution, parameters are assumed to be a function $h_i$ of the vector of time-varying climate

covariates $\boldsymbol{x(t)}$. In the general case, Eq. (1) takes the form:

$$Y(t) \sim GEV(\boldsymbol{\theta}(t)) \tag{4}$$

with $\boldsymbol{\theta}(t) = (\theta_1(t), \theta_2(t), \dots, \theta_m(t))$ the collection of $m$ distribution parameters at time $t$, and

$$\theta_i(t) = h_i(\boldsymbol{x}(t); \boldsymbol{\beta_i}) \qquad i = \{1, 2, \dots, m\} \tag{5}$$

Here $\boldsymbol{\beta_i}$ is the vector of (internal) parameters used in function $h_i$ (not to be confused with parameters $\theta_i$).

The climate-informed GEV is a generalization of the classical GEV. The likelihood function is then defined as:

$$f(\mathbf{Y}|\boldsymbol{\theta}) = \prod_t f(Y(t)|\theta(t)) = \prod_t f(Y(t)|h_1(\boldsymbol{x}(t), \boldsymbol{\beta_1}), h_2(\boldsymbol{x}(t), \boldsymbol{\beta_2}), \dots, h_m(\boldsymbol{x}(t), \boldsymbol{\beta_m})) \tag{6}$$

The function $h_i$, linking the distribution parameters with climate covariates, is derived by means of a linear regression. The shape parameter is assumed to be constant as its estimation includes large uncertainties, even under the assumption of stationarity (Coles, 2001, Papalexiou and Koutsoyiannis, 2013; Silva et al., 2017). A preliminary analysis considering the

150 effect of a covariate on both the location and scale parameter (cf. section 2.3 below) did not provide very different results than those for a covariate on the location parameter only (not shown). Consequently and for reasons of parsimony, we examine only conditional extreme value distributions with a time-varying location parameter.

Conditional distributions of only one covariate at a time are derived, since we are interested in the separate effect of each individual climate index on flood quantiles. Based on the above mentioned assumptions concerning model structure and the

155 form of the function $h_i$, Eq. (5) can be simplified to:

$$\mu(t) = \mu_0 + \mu_1 x(t) \tag{7}$$

where $\mu(t)$ is the varying location parameter, $\mu_0$ the location intercept, $\mu_1$ the location slope and $x(t)$ the single covariate examined.

Consequently, the conditional GEV comprises four parameters: scale and shape parameters, and intercept $\mu_0$ and slope $\mu_1$ for

the location parameter. Since five different climate covariates $x(t)$ are investigated, we construct six different models (one classical and five conditional) for each station and season. The posterior pdf of parameters in Eq. (2) for both the classical and conditional model is estimated using a No-U-Turn Sampler (NUTS) - Hamiltonian Monte Carlo (HMC) approach implemented in Rstan, the R interface to Stan (Stan Development Team, 2017). NUTS is an extension to HMC, a Markov chain Monte Carlo (MCMC) algorithm that avoids the random walk behavior and sensitivity to correlated parameters which characterize

many MCMC methods (Hoffman and Gelman, 2014). Stan is a state-of-the-art platform for statistical modelling and high-performance statistical computation.

For all covariates and seasons, models are fitted independently. No posterior distributions from the classical approach are used as priors for the climate-informed case. For all models, non-informative uniform priors are used for the location parameter (for both intercept and slope) and for the scale parameter, since no prior information is available. For the shape parameter an informative normal distribution with mean 0.093 and standard deviation 0.12 is used. This distribution is adopted from a global study of extreme rainfall by Papalexiou and Koutsoyiannis (2013), which, to our knowledge, summarizes an analysis of shape parameters using the largest number of stations with hydrological data worldwide. Although rainfall extremes may be characterized by slightly different shape parameter than those of streamflow, our informative prior is very close to the "geophysical prior" of Martins and Stedinger (2000), which is often used to restrict the range of shape parameters based on previous hydrological experience (Renard et al. 2013). The latter prior was not preferred because it is bounded to the interval (-0.5, 0.5), while the distribution of Papalexiou and Koutsoyiannis (2013) allows more extreme shape values with a low probability.

Five chains of 14,000 simulations, with the first half discarded as warmup period, are run for all parameters. Convergence is investigated by the potential scale reduction statistic $\hat{R}$ (Gelman and Rubin, 1992). Following Gelman (1996), we assume convergence for values of $\hat{R}$ below 1.2. Thinning is applied to the post-warm up simulations to remove autocorrelation. Every tenth value from all chains is kept, leading to a final sample of 3,500 simulations for all each model and season.

### 2.3 Model selection

We apply a two-step methodology to select the optimal model among the classical and conditional competitors. First, we assess if the covariates have a significant effect on our extreme streamflow models by examining the posterior distribution of the slope $\mu_1$ of the location parameters (Eq. 7). Conditional models are considered as significant if the zero value is not included in the 90% posterior interval of the slope parameter (and thus not by means of a significance test). A second criterion is additionally adopted in order to select the distribution with the best performance by taking into consideration that complex models with more parameters tend to fit the data better. The Deviance Information Criterion (DIC) (Spiegelhalter et al., 2002) is chosen for model selection. The DIC was preferred against two more common tools, the Akaike Information Criterion (AIC; Akaike, 1974) and the Bayesian Information Criterion (BIC; Schwarz, 1978), because it is based on the posterior distribution of the model parameters and thus includes parameter uncertainties, while the AIC and BIC are based on maximum likelihood estimates of parameters.

The deviance, used for the calculation of the DIC, is defined as:

$$D(\boldsymbol{\theta}) = -2log(f(\boldsymbol{Y}|\boldsymbol{\theta})) \tag{8}$$

where $\boldsymbol{\theta}$ is the parameter vector. The DIC is then given by the following equation:

$$DIC = \bar{D} + p_D \tag{9}$$

where $\bar{D}$ is the expectation of the deviance with respect to the posterior distribution, and $p_D = \bar{D} - D(\bar{\boldsymbol{\theta}})$ is the effective number of parameters (penalty for model complexity, following Spiegelhalter et al., 2002). $\bar{\boldsymbol{\theta}}$ is a vector of the expectation of parameters $\boldsymbol{\theta}$. Models with smaller DIC values are preferred.

Conditional models satisfying both criteria are preferred to the classical model. The model comparison is performed in two steps: first, for each station and season, each climate-informed competitor is pairwise compared to the classical GEV. Subsequently, the model with the overall best performance is identified.

### 2.4 Conditional flood quantiles

In the classical or stationary approach one can define the n-year return level as the high quantile of the examined variable for which the probability of exceedance is 1/n. In this case, the same probability of exceedance is assigned to same events in different years. The concept of return period can then be introduced as the reciprocal of the probability of exceedance of a specific value or return level of the examined variable (Cooley, 2013). In engineering practice, return period is often used to

communicate risk and is understood either as the expected time interval at which the examined variable exceeds a certain threshold for the first time (average occurrence interval) or as the average of the time intervals between two exceedances of a given threshold (average recurrence interval) (Volpi et al., 2015). When the parameters of the distribution vary in time, as in the nonstationary or conditional frequency analysis, a different probability of exceedance is assigned to different years. In this case, the concept of return period becomes less straightforward to define. Thus, communicating risk by means of probabilities makes more sense (Cooley, 2013). Instead of the classical return levels the term "effective" return levels has been introduced (Gilleland and Katz, 2016) which represents the quantiles of the conditioned distribution under consideration of a particular value of the covariate during a given year.

Here we assess whether the consideration of climatic drivers leads to a significant alteration of flood "effective" return levels or conditional quantiles in individual years. Differences of flood quantiles during years with high and medium values of the considered circulation indices are quantified. Since the model is linear, the effect of high and low covariate values on the extreme value distribution quantiles is approximately symmetric (it would be symmetric if the seasonal indices had a symmetric distribution around zero – see Fig. S6) and thus low covariate values are not considered. The $95^{th}$ and $50^{th}$ quantile of the considered climate index are chosen as high and medium index values, respectively. Index quantiles are calculated for the entire period 1950-2016.

From the No-U-Turn sampling after thinning, 3,500 post-warm up sets of parameters are obtained, each corresponding to a flood quantile (for a given probability of exceedance). The median value of all 3,500 flood quantiles is chosen as a point estimate. The median estimate was preferred to the maximum a posteriori (MAP) estimate because it is more representative of the posterior distribution. Based on this approach, the percent relative difference $Y_p$ of the two flood quantiles for a particular probability of exceedance $p$, corresponding to the high and medium climate index quantiles, respectively, is calculated as follows:

$$Y_p = \frac{y_{p,h} - y_{p,m}}{y_{p,m}} (\%) \tag{10}$$

where $y_{p,h}$ is a flood quantile for the probability $p$, incorporating a high value of the considered climate index ($95^{th}$ quantile). $y_{p,m}$ is the quantile value for the same probability $p$ under consideration of the medium ($50^{th}$ quantile) climate index. The analysis is performed for probability of exceedance of 0.02 (corresponding to the 50-year return period of the classical case).

## 2.5    Uncertainty analysis

In the previous chapters an automatic methodology for the choice of an adequate model and a discussion of flood quantiles for different covariate values is presented. However, a visual comparison of point estimates and uncertainty intervals of the classical and conditional models can be useful, since it illustrates the differences but also the plausibility and possible drawbacks of the competing models. For this reason, we plot the time series of flood quantiles for a probability of exceedance of 0.02 for selected gauges and covariates based on both the classical and the climate-informed extreme value distribution. As discussed in the previous section, the median flood quantile for a probability of exceedance of 0.02 is chosen as point estimate (median quantile curve). Uncertainty of flood quantiles is quantified by means of posterior or credibility intervals, which are the Bayesian equivalent to frequentist confidence intervals, although there exist differences in the interpretation of the two types (Renard et al. 2013, Gelman et al., 2013).

## 3.    Results

### 3.1    Spatial patterns of competing models

For all seasonal indices climate-informed models are preferred over the classical distribution for a large number of stations; percentages of preferred models (based on both the DIC and the significance of the slope of the location parameter) are shown

in Table 1 and spatial patterns are mapped in Fig. 1-2. The climate-informed fits form spatial clusters that resemble the correlations between the climate indices and average seasonal precipitation (Fig. S1-S4), while a relation with the correlations of seasonal mean temperature is not straightforward. Particularly for NAO a dipole pattern is evident in winter, with a positive influence on extreme discharge in northern and Central Europe and a negative relationship south of the Alps (Fig. 1). The intra-annual shift of the NAO pressure centers is well captured. The positive influence of NAO on flood magnitudes during summer is only detected for northern Scandinavia (Fig. 2). Similar dipole structures, resembling the correlations with seasonal mean precipitation, are found for other indices. However, there are some deviations from the precipitation patterns. For example, contradicting results are found in Scandinavia during spring and summer for the SCA index. Scandinavian rivers usually have small catchments and are particularly fed by snowmelt in spring, subsequently in this area, both temperature and precipitation are important for runoff generation. An opposite sign between correlations with precipitation and the slope of the location parameter can also be found during autumn in north-eastern Germany for the EA index.

NAO is the covariate with the highest number of significant fits in winter (46%) and autumn (31%) and EA in spring (32%) and summer (18%). High percentages of preferred climate-informed models are also found for EA and SCA in winter, which is the season where most indices are characterized by their strongest influence on the European climate (Table 1). Worst overall results are found for EA/WR in spring (3%) and POL in summer (7%). It can be argued that these two latter cases could occur solely by chance or due to spatial correlation of nearby flood time series; however, results are coherent in space and cover large regions, which suggests a real influence of the circulation modes on the location parameter of the extreme value distributions, restricted though to certain sub-regions of Europe.

Similar spatial patterns are obtained from the same analysis if monthly covariates during the month of the seasonal discharge peaks are examined (Fig S7-S8). Clusters of stations with positive or negative slopes of the location parameter agree with those for seasonal indices, however in most cases the percentages of preferred fits are lower for the monthly covariates, with EA/WR in spring being an exception. In particular, the role of NAO in winter and autumn and of EA during the rest of the seasons is less pronounced in the monthly-scale analysis. NAO and SCA are the covariates with the highest number of preferred fits in spring and EA during the rest of the seasons, together with EA/WR in summer (Table 2). Regarding the spatial patterns of preferred fits, deviations from those for seasonal covariates can be found for EA/WR, SCA and POL during spring and summer.

For all indices examined, a percentage of stations between 5 and 13%, depending on the season and the covariate, are characterized by lower DIC for the climate-informed model although the slope of the location parameter is not significant (illustrated as yellow points in Fig. 1 and 2). Only a few station records, up to three per season and index (not shown in Fig. 1 and 2), are characterized by higher DIC value for the climate-informed model without showing a significant slope. These results indicate that DIC is a weaker criterion for model selection than the slope significance at 10% level.

In order to illustrate the spatial structure of best models, the preferred model (classical or climate-informed) is mapped in Fig. 3 and 4 for each station for seasonal covariates. Spatial patterns do not resemble the pattern of significant fits for separate indices (Fig. 1, 2), since the influence of the selected climate modes on flood frequencies is overlapping for some regions and some of the indices are correlated for particular seasons (Table S1). Winter (summer) is the season with the highest (lowest) overall percentage of preferred climate-informed models: 77% and 38%, respectively. In winter, NAO is the most influential climate mode, being preferred over the other modes for 28% of the gauges. The largest influence of NAO on flood frequencies is detected in Central Europe, Great Britain, parts of Scandinavia and the Iberian Peninsula (Fig. 3). The first three regions show also a high fraction of SCA-influenced models, which points towards a joint effect of NAO and SCA during winter. The two indices are significantly correlated during this season (Table S1). EA is identified as the best covariate in winter for Great Britain. In spring an expansion of the EA influence towards Central Europe is detected. The NAO influence is shifted to the south during the transition seasons (spring and autumn) and is completely dissolved in summer. Patterns for SCA are

heterogeneous throughout the year. The same results but for monthly covariates are shown in Fig. S7 and S8. Spatial patterns resemble those for seasonal covariates. Percentages of preferred climate-informed models are included in Tables 1 and 2.

### 3.2     Conditional quantiles and uncertainty analysis

In the previous section it is shown that models with monthly covariates do not outperform those with seasonal covariates for most indices and seasons. Hence, quantiles of climate indices are calculated at the seasonal scale only (Table 3). Figures 5 and 6 show the relative differences of seasonal flood quantiles for a probability of exceedance of 0.02 between a (hypothetical) year with a climate index value equal to the 95[th] index quantile and a year with an index value equal to the median. For a probability of exceedance of 0.02, relative differences higher than 20% and up to 22% are detected in winter for NAO. For the rest of the seasons, maximum relative differences are lower than 20% with highest values for EA/WR in autumn (marginally below 20%). In spring and summer the highest value is considerably lower, between 11-13% for NAO and SCA in spring and EA and SCA in summer.

A difference of 5-10% is quite common for NAO in winter. For example, a station with a positive slope of the location parameter and a probability of exceedance of 0.02 for a maximum seasonal discharge value of 600 m$^3$/s during years characterized by a medium NAO index has an effective return level between 630 and 660 m$^3$/s during years with a highly positive NAO state. Particularly for Great Britain and Scandinavia, high relative differences, positive or negative, are found in winter for different indices. Differences of extreme discharge higher than 10% are characteristic for variations of the EA index in south-eastern Britain and for EA/WR in Norway. Some stations with high differences are also found in Norway and northern Britain for NAO and SCA in spring. Summer is characterized by low relative differences, below 5% for most stations. On the contrary, in autumn clusters of stations with medium to high differences, positive or negative (higher than 5% and locally exceeding 10%), are found in Scandinavia for NAO and EA/WR, in entire northern Europe for EA and in the Alpine region, southern Great Britain and Norway for SCA.

The high relative differences of flood quantiles could partly reflect differences in catchment size or unreasonable posterior values of the shape parameter. A link with catchment size was, however, not found (not shown). Posterior shapes for all seasons and indices were further analyzed. Summary statistics of the median shape from the posterior distribution of each fitted model are given in Table 4. Little deviation is observed for different models (classical or climate-informed) during the same season but some inter-season variation is present. No unreasonable values are observed, thus we assume that the use of an informative prior distribution for shape adequately restricts the posterior distributions to reasonable limits.

The results for three selected gauges with high relative differences $Y_{0.02}$ are presented in detail. The selected stations cover different characteristic combinations with regard to the investigated season and the considered covariate. The time series of discharge values with a probability of exceedance of 0.02 are illustrated for the classical case and the climate-informed case for the three indices with the lowest DIC (Fig. 7). Conditional quantiles are calculated on a year-to-year basis, based on the observed values of the selected climate indices. Details about the streamflow gauges and the climate-informed fits are given in Table 5 and 6, respectively. Results show that the conditional and unconditional point estimates and uncertainty bounds can differ considerably, particularly for models with a high relative difference $Y_{0.02}$ and a low DIC (subplots A1, B1 and C1 in Fig. 7). Obviously results from the conditional models vary with time. For example, for the station Asbro 3 in Sweden, strongly different results are obtained by the classical and the NAO-conditional model in winter, particularly for the period 1960-1970, which was dominated by negative NAO conditions and reduced winter precipitation amounts over Northern Europe. The same applies for the station Teston in Great Britain during the period 1960-1980, if EA is considered as a covariate. These results show that the climate-informed models can modulate the estimated flood risk for single years or longer periods and thus substantially deviate from the estimation based on the classical distributions. For models characterized by small relative differences or insignificant slopes of the location parameter (subplots A3, B3 and C3), conditional uncertainty bounds tend to

converge to a straight line resembling the classical case. The classical case is theoretically a subcase of the climate-informed model. However, the two models are fitted independently and the two intervals do not always overlap.

The uncertainty bounds of the climate-informed fits can be narrower or wider than those of the classical model. They are also asymmetric, contrary to uncertainty bounds that result from a method using a normal approximation. Asymmetric intervals are associated with the shape parameter of the GEV and are not uncommon (see for example Zeng et al., 2017). The range of uncertainty bounds reflects an interplay between model complexity and the additional information provided by the more complex models. In Fig. 7, uncertainty bounds are narrower in the case of the "best" conditional models (e.g. subplot A1). Uncertainty increases when extrapolations are made towards high and low index values. This can be more easily observed in Fig. 8. For the classical case, the range is about 94 m$^3$/s. For the climate-informed case and NAO = 0 (close to its median value) the range is around 70 m$^3$/s. The range increases to 74 m$^3$/s for NAO = 1 and to 80 m$^3$/s for the most extreme observed NAO value (NAO = -2.1). For a NAO value around 3/-3 the range of uncertainty bounds reaches that of the classical model.

## 4.    Discussion and conclusions

This study explored whether a climate-informed flood frequency analysis provides insights and can improve the estimation of flood probabilities at the European scale. A site-specific model using a Bayesian framework was developed, and five Euro-Atlantic circulation modes were investigated as potential covariates: the North Atlantic Oscillation (NAO), the East Atlantic pattern (EA), the Scandinavia pattern (SCA) and the Polar/Eurasian pattern (POL). Streamflow was analyzed at a seasonal time scale in order to account for the variable influence of the circulation modes on the European climate during different seasons of the year. Covariates were averaged and examined at both seasonal and monthly scales, contemporaneous to the season or month of the seasonal streamflow maxima, respectively.

The developed climate-informed models were compared to the classical GEV with time-invariant parameters. For most seasons and covariates investigated, the climate-informed models were preferred over the classical GEV for a high percentage of stations (around 20% on average), with best results found in winter for NAO and EA, in spring for EA and in autumn for NAO (Table 1). Results were shown to be coherent in space, indicating that certain regions are influenced by particular circulation modes (Fig. 1-4). In winter 77% of the stations were found to be influenced by one of the climate modes which indicates a high potential for an improvement of flood probability estimations by including climate information into extreme value statistics. On the contrary, less than half of the stations examined were significantly affected by at least one of the five large-scale indices during summer season, indicating a rather convective and non-predictable precipitation regime (Table 1).

Based on the variability of the circulation indices, we identified regions that are characterized by preferred climate-informed fits and by steep slopes of the location parameter. For models with significant slopes, variations of the climate indices lead to highly varying flood quantile estimations for the same probability of exceedance. Particularly for northwest Scandinavia and the British Isles, variations of the climate indices result in considerably different extreme value distributions and thus highly different flood estimates for individual years (Fig. 5-6). This difference in estimates could be partly a result of unreasonable posterior values of the shape parameter, however, the use of an informative prior distribution for shape adequately restricts the posterior distributions to reasonable limits. Plots of extreme streamflow under consideration of a probability of exceedance of 0.02 indicate that the deviation between the classical and climate-informed analysis concerns not only single years but can also persist for longer time periods (Fig. 7), which reflects the decadal-scale variability of NAO and other large-scale circulation indices (Fig. S5).

Although the circulation indices examined are characterized by a high intra-seasonal variability, the seasonally averaged indices provided in most cases better fits compared with monthly values (Tables 1-2). This should be emphasized, since extreme precipitation events are most likely stronger related to monthly circulation states, which better represent the moisture fluxes into the target domain. On the contrary, the catchment wetness before the flood event is likely to be influenced by the

seasonal mean circulation and the associated precipitation sums. Hence, our result suggests that the skill of climate informed extreme values distributions is to a significant part a consequence of the important link between catchment wetness and flooding. Thus we assume, in line with recent studies (Blöschl et al., 2017; Merz et al., 2018; Schröter et al., 2015), that in many regions of Europe, catchment wetness plays an important role for flood generation.

For the selection of the best model among the classical and climate-informed two criteria were adopted: the DIC and the significance of the slope of the location parameter $\mu_1$. For all indices and seasons, the DIC favored the climate-informed models over the classical distribution for a larger number of stations compared to the slope significance. DIC has received some criticism for not adequately penalizing complex models and tending to choose overfitted models (Silva et al., 2017; Spiegelhalter et al., 2014). Our results show that at least compared to the slope significance, DIC is a weaker criterion for 380 model selection. A criterion comprising a higher penalty term for model complexity could alternatively be adopted. A more conservative version of DIC has been proposed by Ando (2011) but is not commonly used until today (Silva et al., 2017). The described methodology can be complemented in several ways.

(a) Regional framework

In this study, a local, site-specific flood frequency model was developed. This model allowed to identify spatial coherence in 385 relations between streamflow extremes and large-scale atmospheric patterns. However, a shortcoming of this methodology is the high uncertainty of streamflow estimates for high probabilities of exceedance (corresponding for example to the 100- or 200-year flood). Instead of a local framework, a regional framework can be alternatively implemented. The latter, by considering all available streamflow information in a region, decreases uncertainty and offers the possibility of improving streamflow quantile estimation.

(b) Alternative models

A linear relationship was assumed between streamflow extremes and the large-scale atmospheric indices. This is a simplification of reality and some relations may be over- or underestimated due to existing non-linearities in the climate-streamflow system. More complex, particularly non-linear models would also be possible candidates for describing the relation between climate indexes and flood probabilities. However, with increasing model complexity, the chances for model 395 overfitting also increase. In this study we assumed a symmetric influence of the positive and negative phases of the climate indices. However, an asymmetric relation may better describe the effect of certain climate modes on streamflow extremes. For example, Sun et al. (2014) used an asymmetric piecewise-linear regression to account for the different effects of El Niño and La Niña on rainfall extremes in Southeast Queensland, Australia. Furthermore, we also assumed a varying location parameter and constant scale parameter. A constant coefficient of variation as in Serago and Vogel (2018) would also be possible and as 400 parsimonious as our model. In this case, a varying scale parameter linked to the location parameter would need to be implemented.

(c) Number of covariates

Single covariate models were developed, focusing on the separate effect of each individual climate mode. The methodology can be extended to a model considering several covariates at the same time. In that case, dependencies between the covariates, 405 if existent, should be taken into consideration. López and Francés (2013) overcame this problem by using the principal components of climatic indices as covariates for the flood frequency analysis. This, however, increases the model complexity considerably and thus the chances of model overfitting. This needs to be considered in developing models with multiple covariates.

(d) Contemporaneous and lagged relationships

In this paper we considered contemporaneous relationships between streamflow extremes and pressure modes that directly shape the European climate and hydrology. However, lagged relationships may prove more useful for flood risk management and the (re-)insurance industry, since they would allow forecasts of temporal variable flood quantiles for the following month or season. The contemporaneous streamflow-covariate setup presented here can be used, together with a seasonal prediction

of indices, for an ahead-season forecast of streamflow quantiles. In this case covariate uncertainty must be additionally considered. A second possibility is to operate the presented model in a forecast mode under consideration of different time lags between selected covariates and observed streamflow maxima. Our results suggest that catchment wetness has an important role in shaping seasonal maximum streamflow. In a follow up study, we will systematically test the skill of various predictor variables, describing both the climate and catchment state, in forecasting runoff extremes in Europe.

**Data availability**

The GRDC discharge dataset was obtained from The Global Runoff Data Centre, 56068 Koblenz, Germany (https://www.bafg.de/GRDC/EN, last access in October 2017) and is available upon request. Time series of monthly circulation indices were retrieved from the Climate Prediction Center (CPC) of the National Oceanic and Atmospheric Administration (NOAA) and can be accessed through http://www.cpc.ncep.noaa.gov/data/teledoc/telecontents.shtml. Additional discharge data from Spain and Portugal were provided upon request by Luis Mediero and for Pontelagoscuro, Italy by Alessio Domeneghetti. Gridded pressure data were extracted from the NCEP/NCAR Reanalysis dataset and are provided through http://www.esrl.noaa.gov/psd/. Gridded temperature and precipitation data were extracted from the CRU TS3.24 dataset from the climatic research unit (CRU, https://crudata.uea.ac.uk/cru/data/hrg/) of the University of East Anglia.

**Author contributions**

B.M. conceived the original idea, and all co-authors designed the overall study. E.S. developed the model code with contributions from X.S, performed the analysis, and prepared the manuscript. All co-authors contributed to the interpretation of the results and writing of the manuscript.

**Acknowledgements**

The authors are grateful to the three reviewers, Alberto Viglione, Elena Volpi and Francesco Marra for their helpful comments and suggestions that substantially improved the manuscript. Alessio Domeneghetti is thanked for providing unpublished discharge data from Italy and Luis Mediero for providing discharge data from Spain and Portugal. Daniel Beiter is thanked for his support in coding and parallel computing. Xun Sun is supported by the National Key R&D Program of China (No. 2017YFE0100700) and Shanghai Pujiang Program (No. 17PJ1402500). This study was conducted in the frame of the projects "Conditional flood frequency analysis: exploring the link of flood frequency to catchment state and climate variations" and "The link of flood frequency to catchment state and climate variations", two joined research initiatives between AXA Global P&C and GFZ, Potsdam. The authors wish to acknowledge the AXA Research Fund for financial support.

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

**Figures**

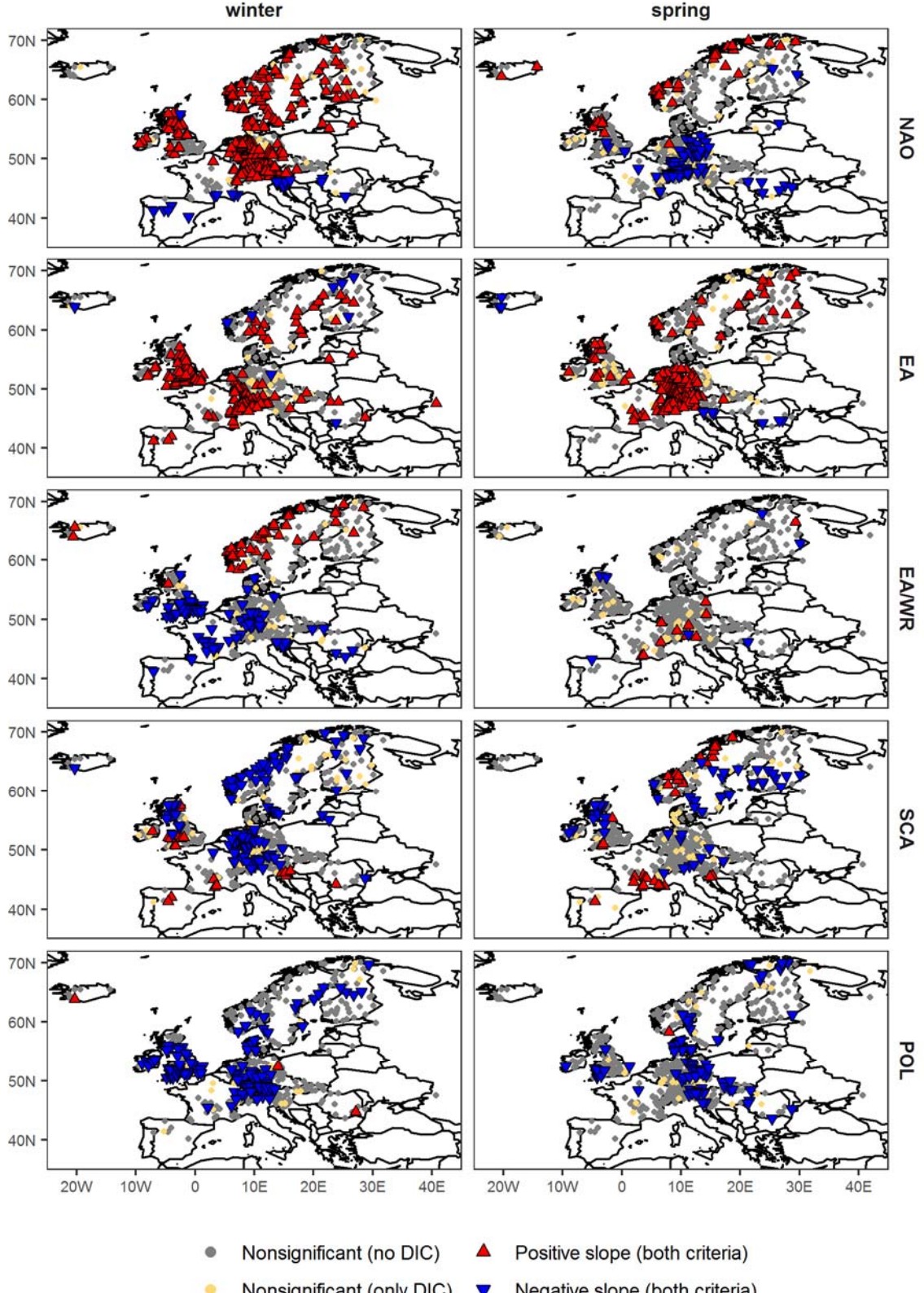

Figure 1: Results comparing the climate-informed and the classical GEV models for all covariates examined for the winter and spring season. Nonsignificant models preferred only by the DIC (yellow points) are plotted on top of stations for which climate-informed models were not chosen by any of the two criteria (grey points). Preferred climate-informed models chosen by both criteria (blue/red triangles) are illustrated on top of the other models so that they can be better distinguished.

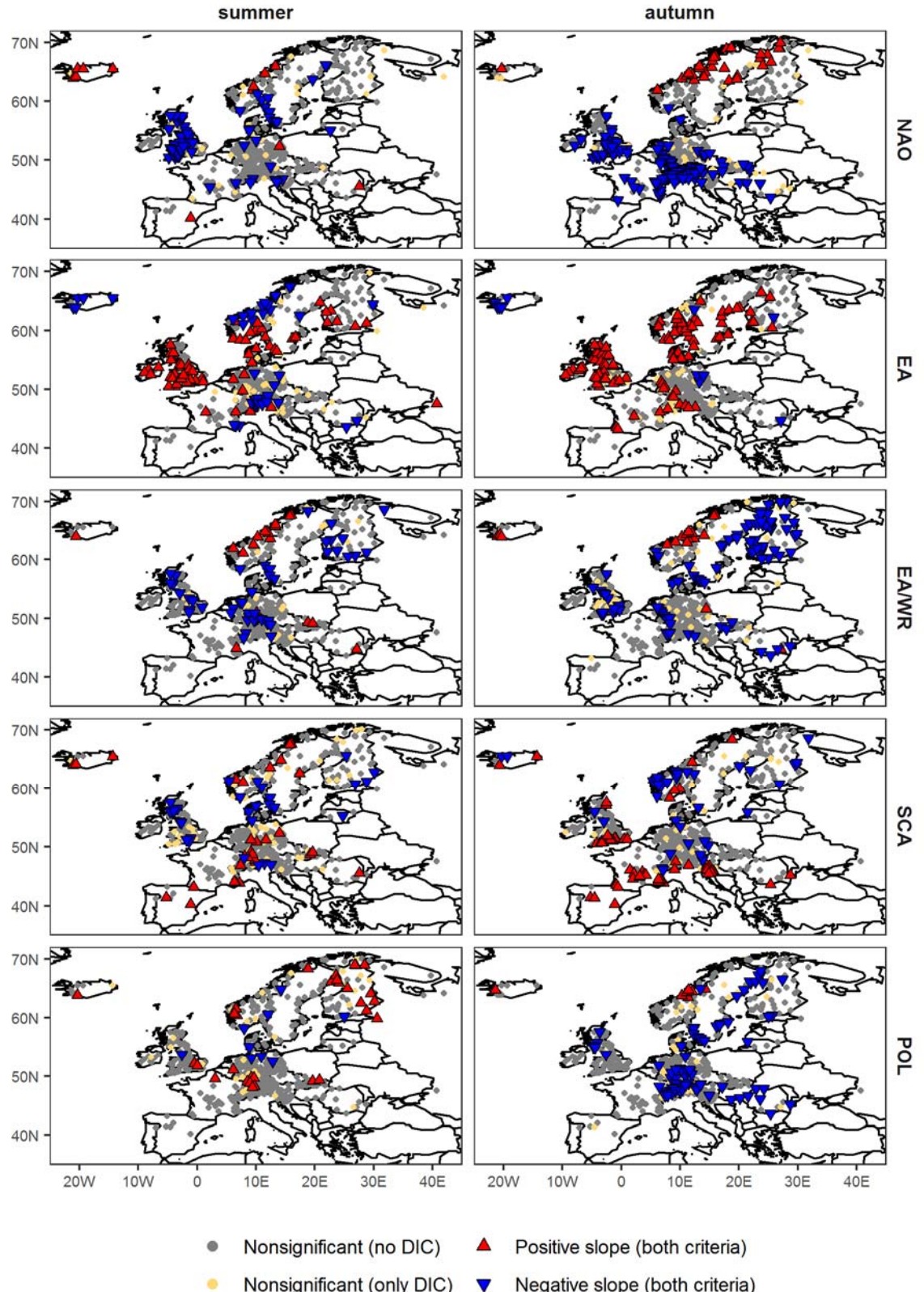

**Figure 2: Same as Fig. 1 but for the summer and autumn season.**

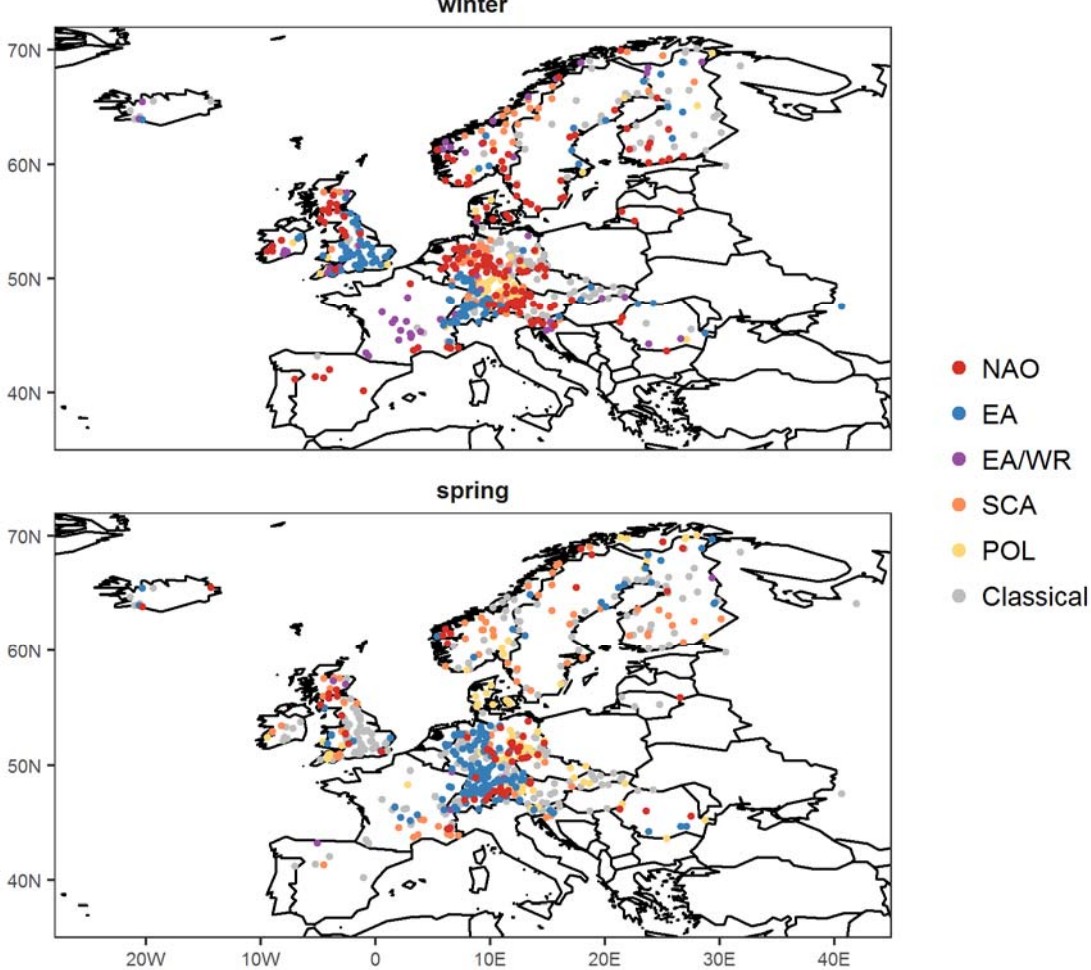

**Figure 3:** Best overall models among the five climate-informed and classical GEV tested for the winter and spring season. Mean seasonal covariates are examined.

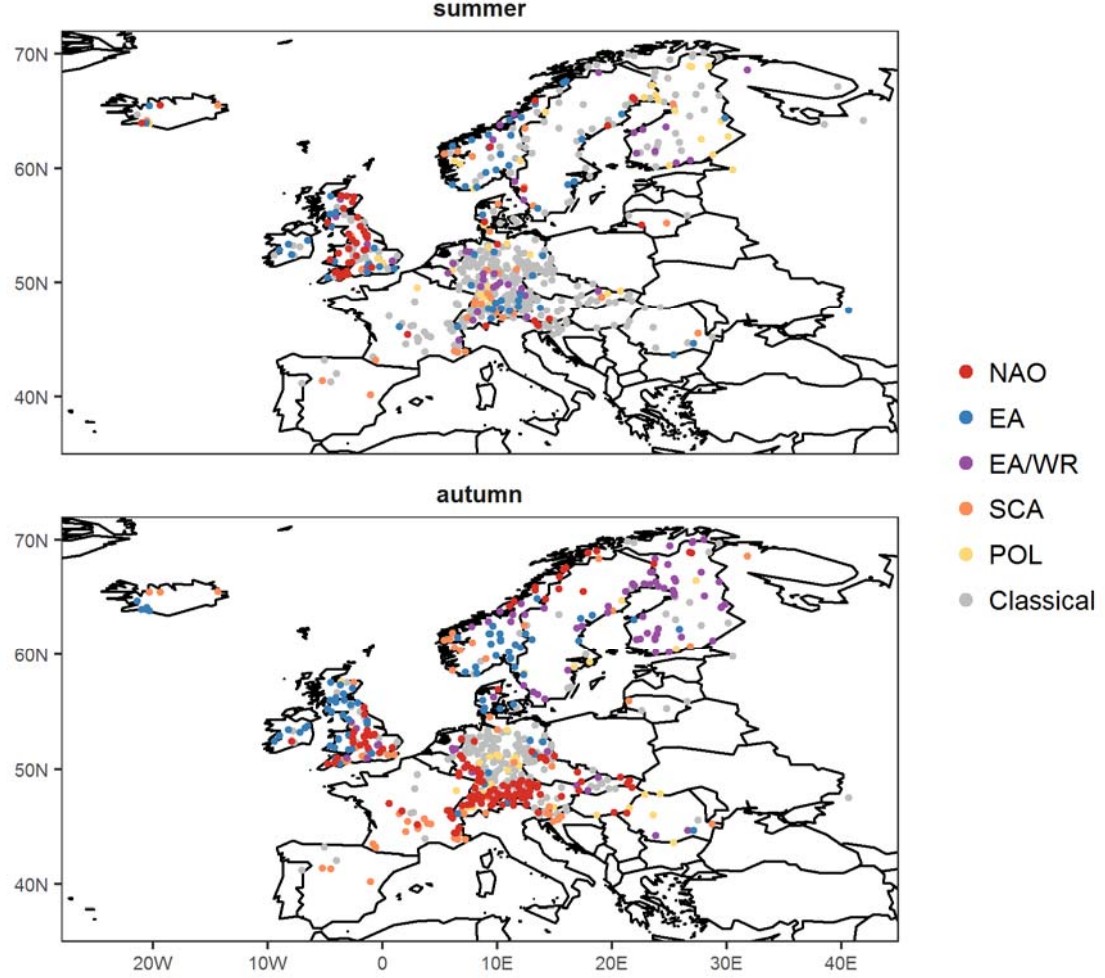

**Figure 4: Same as Fig. 3 but for the summer and autumn season.**

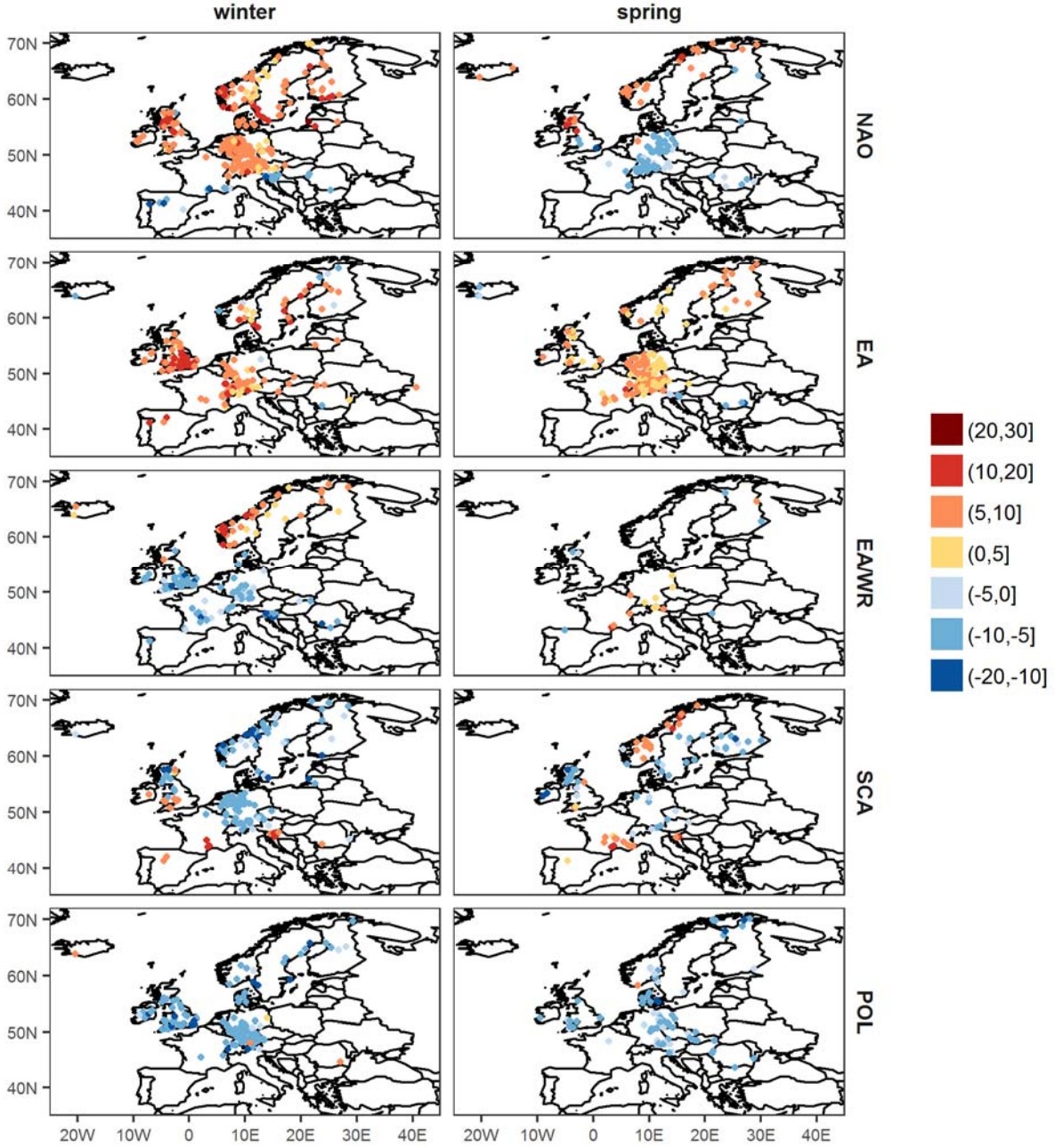

**Figure 5: Percent relative difference of the streamflow for an exceedance probability of 0.02 between a (hypothetical) year with a climate index value equal to the 95th quantile and a year with an index value equal to the median index. Results are shown for winter and spring and seasonal mean covariates.**

600

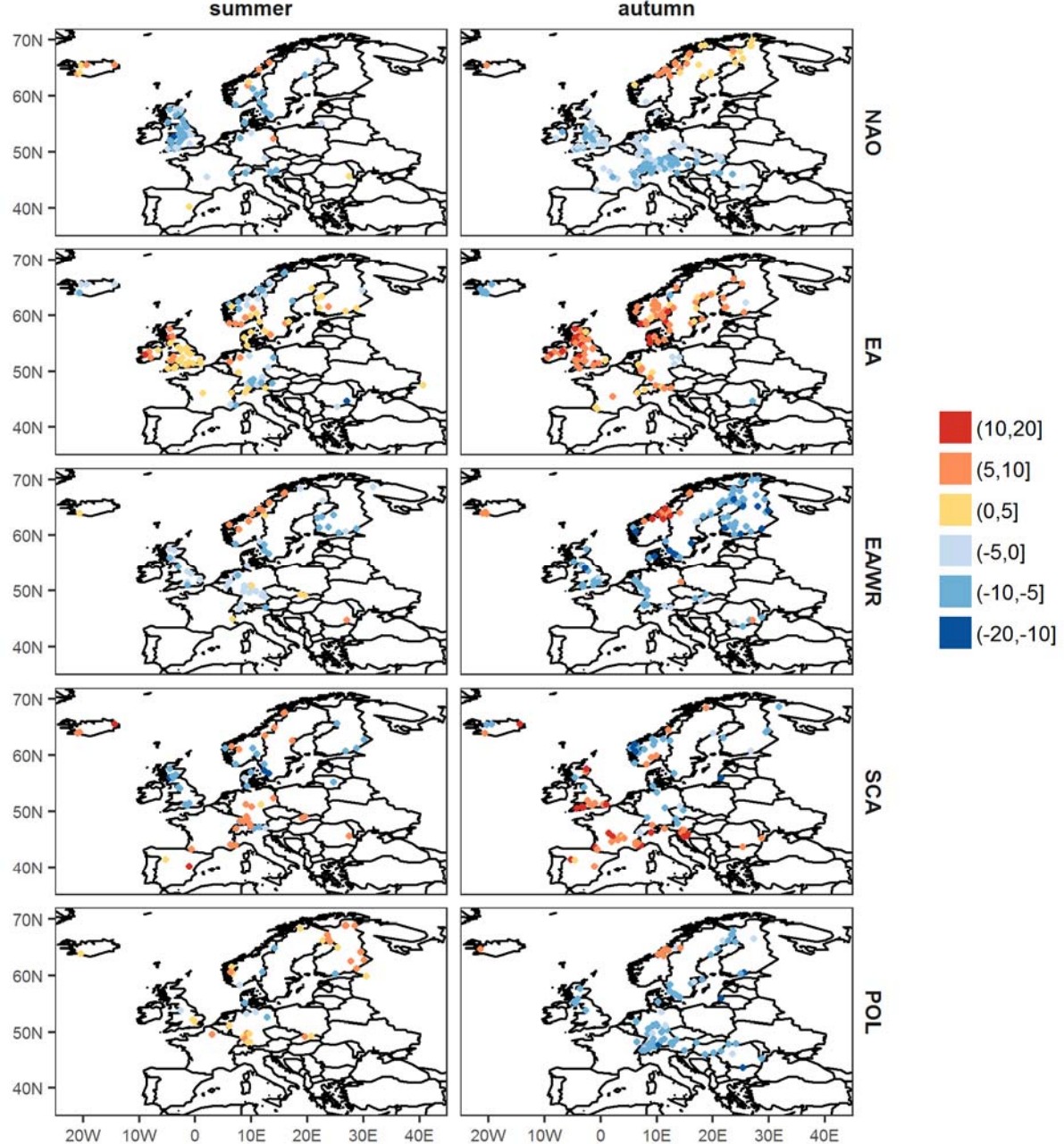

**Figure 6: Same as Fig. 5 but for the summer and autumn season.**

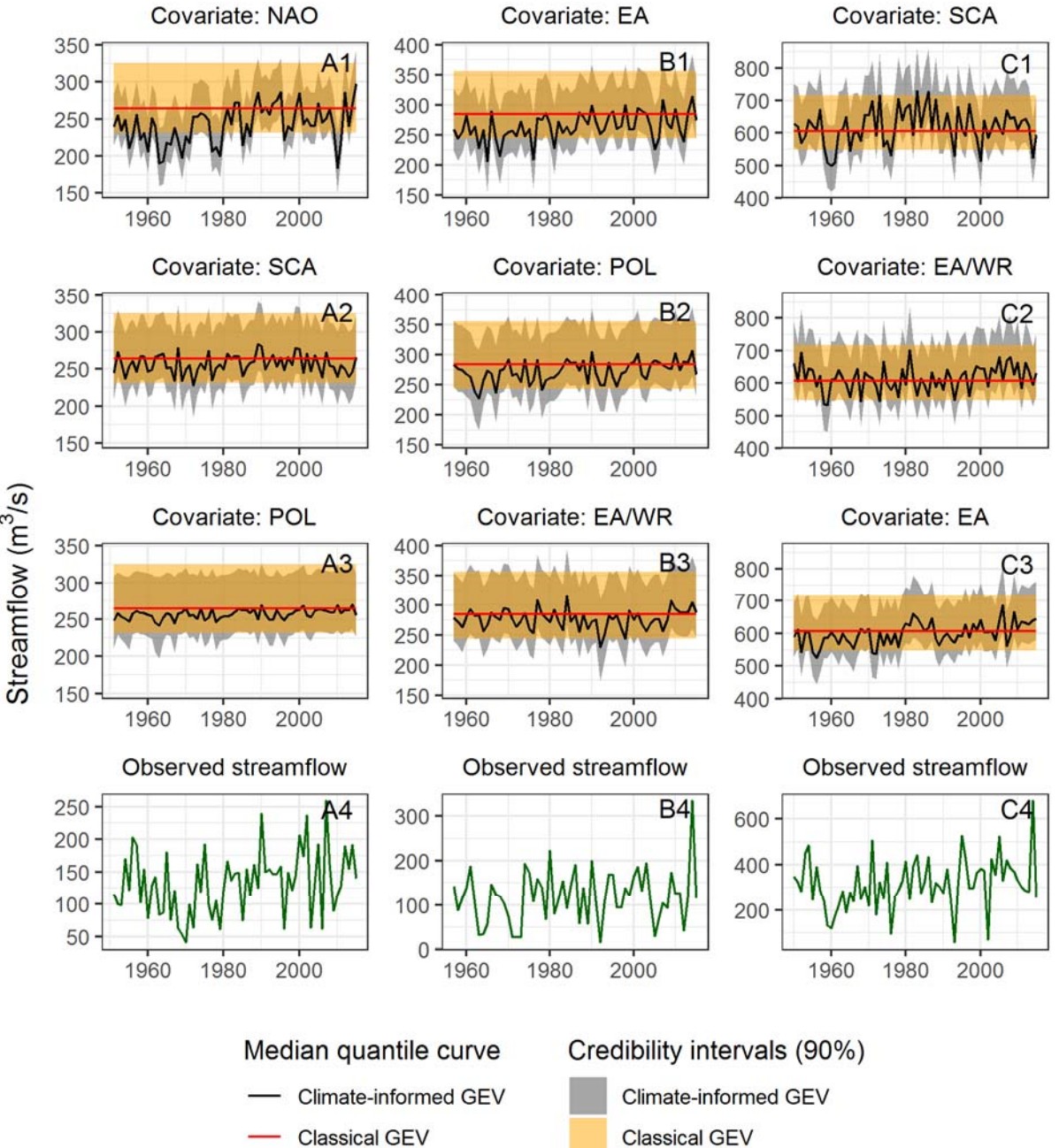

Figure 7: Annual maximum discharge time series (lower panel: 4) and climate-informed quantiles (upper panels: 1-3) with credibility intervals for an exceedance probability 0.02 and for three selected gauges (Table 5, 6). Climate-informed quantiles are compared with those of the classical GEV. The three best climate-informed models based on the DIC are shown for each site, with increasing DIC from top to bottom.

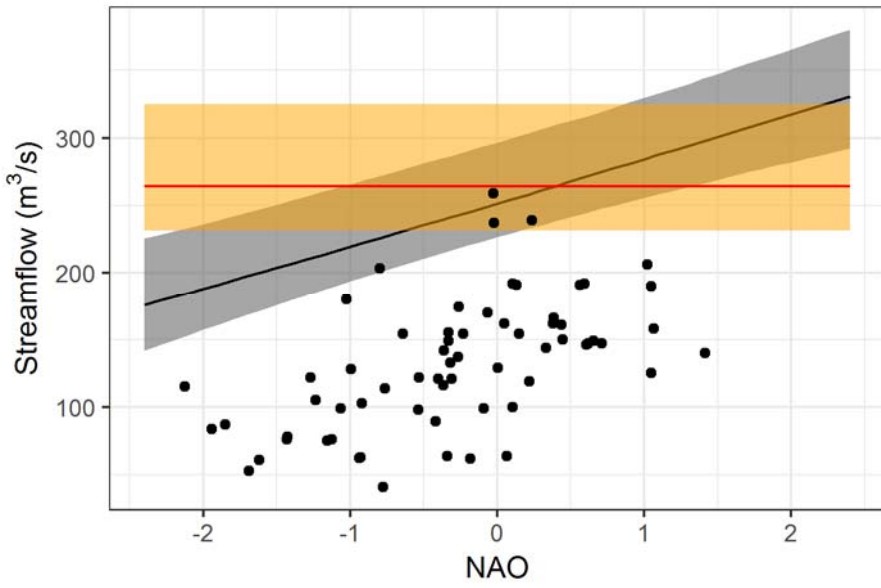

**Figure 8: Comparison of climate-informed and classical streamflow quantiles for station A (see Fig. 7 and Tables 5, 6 for more details), an exceedance probability 0.02 and NAO as covariate of the climate-informed model. The legend is the same as in Fig. 7. Observed streamflow is indicated with black dots.**

615

**Tables**

Table 1: Percentage of stations with climate-informed fits preferred to the classical GEV model. Indicated is the result of the pairwise comparison of each covariate with the classical model and the percentage of preferred fits for each covariate when all models are compared (in brackets). Results are shown per season and for mean seasonal covariates.

| Index | Winter | Spring | Summer | Autumn |
|---|---|---|---|---|
| NAO | 46 (28) | 19 (10) | 13 (8) | 31 (24) |
| EA | 29 (20) | 32 (26) | 18 (11) | 19 (13) |
| EA/WR | 24 (10) | 3 (1) | 10 (7) | 20 (13) |
| SCA | 26 (11) | 14 (12) | 10 (6) | 15 (11) |
| POL | 24 (9) | 14 (11) | 7 (6) | 13 (5) |
| All indices | (77) | (60) | (38) | (66) |

Table 2: Same as Table 1 but for monthly covariates at the same month as the seasonal streamflow extremes.

| Index | Winter | Spring | Summer | Autumn |
|---|---|---|---|---|
| NAO | 33 (28) | 16 (12) | 13 (9) | 13 (8) |
| EA | 27 (18) | 15 (12) | 15 (10) | 22 (18) |
| EA/WR | 26 (14) | 14 (9) | 13 (11) | 17 (12) |
| SCA | 15 (8) | 17 (12) | 12 (9) | 13 (9) |
| POL | 7 (4) | 16 (13) | 6 (4) | 10 (7) |
| All indices | (72) | (58) | (43) | (54) |

Table 3: Seasonal quantiles of the five climate indices: median and in the parenthesis the 95$^{th}$ quantile are provided.

| Index | Winter | Spring | Summer | Autumn |
|---|---|---|---|---|
| NAO | -0.26 (1.04) | -0.15 (0.84) | 0.02 (1.14) | 0.17 (0.96) |
| EA | -0.37 (1.07) | -0.13 (0.70) | -0.07 (0.80) | -0.19 (0.69) |
| EA/WR | -0.19 (0.78) | -0.04 (0.78) | 0.15 (1.23) | 0.11 (1.29) |
| SCA | 0.21 (1.25) | 0.05 (0.90) | 0.09 (1.33) | 0.21 (1.44) |
| POL | 0.11 (1.44) | 0.07 (0.90) | -0.11 (0.94) | -0.02 (0.91) |

**Table 4: Summary statistics of median posterior shape parameter of all stations examined. Statistics are taken over all models for one season. In the parenthesis the maximum deviation of all the models fitted (classical and climate-informed) is provided.**

|  | Winter | Spring | Summer | Autumn |
|---|---|---|---|---|
| **Min** | -0.420 (0.072) | -0.365 (0.057) | -0.303 (0.058) | -0.303 (0.074) |
| **Q5** | -0.137 (0.013) | -0.104 (0.007) | -0.055 (0.009) | -0.057 (-0.010) |
| **Q25** | -0.008 (0.007) | 0.002 (0.010) | 0.062 (-0.005) | 0.045 (-0.06) |
| **Median** | 0.062 (0.006) | 0.066 (0.006) | 0.165 (0.005) | 0.127 (-0.008) |
| **Mean** | 0.063 (0.003) | 0.066 (0.005) | 0.165 (0.002) | 0.125 (-0.008) |
| **Q75** | 0.133 (-0.006) | 0.133 (-0.005) | 0.271 (-0.005) | 0.200 (-0.010) |
| **Q95** | 0.262 (-0.014) | 0.226 (-0.009) | 0.385 (-0.006) | 0.316 (0.009) |
| **Max** | 0.461 (-0.053) | 0.381 (-0.019) | 0.537 (-0.020) | 0.527 (-0.031) |

**Table 5: General information about selected sites shown in Fig. 7. Ref. code is the number of the subplot of Fig. 7.**

| Ref. code | Station name | Country | GRDC No | Latitude | Longitude | Catchment size (km$^2$) |
|---|---|---|---|---|---|---|
| **A** | ASBRO 3 | Sweden | 6233100 | 57.240 | 12.310 | 2160.2 |
| **B** | TESTON | United Kingdom | 6607851 | 51.251 | 0.447 | 1256.1 |
| **C** | BULKEN | Norway | 6731200 | 60.630 | 6.280 | 1102.0 |

**Table 6: Climate-informed results as shown in Fig. 7. Ref. code is the number of the subplot of Fig. 7. Mean seasonal covariates for the same season as streamflow extremes are examined. dDIC is the difference from the DIC value of the classical distribution. $Y_{0.02}$ is the percent relative difference of streamflow with exceedance probability 0.02 for the 95$^{th}$ quantile of the covariate ($y_{0.02,h}$) and of streamflow with exceedance probability 0.02 for the median ($y_{0.02,m}$). The sign of the slope is reported if it is significantly different than zero at the 10% significance level.**

| Ref. code | Season | Covariate | dDIC | Slope | $Y_{0.02}$ [%] | $y_{0.02,m}$ [m$^3$/s] | $y_{0.02,h}$ [m$^3$/s] |
|---|---|---|---|---|---|---|---|
| **A** | Winter | NAO | -22.9 | Positive | 17.7 | 242.7 | 285.6 |
|  |  | SCA | -4.3 | Negative | -7.9 | 255.3 | 235.2 |
|  |  | POL | -0.5 | Nonsignificant | - | - | - |
| **B** | Winter | EA | -9.4 | Positive | 14.7 | 260.5 | 299.0 |
|  |  | POL | -4.6 | Negative | -12.0 | 273.8 | 241.0 |
|  |  | EA/WR | -4.2 | Negative | -8.5 | 276.8 | 253.3 |
| **C** | Autumn | SCA | -15.5 | Negative | -15.8 | 613.7 | 516.8 |
|  |  | EA/WR | -5.9 | Negative | -11.2 | 615.4 | 546.0 |
|  |  | EA | -3.6 | Positive | 7.7 | 601.6 | 648.0 |