# Peer review of "Climate influences on flood probabilities across Europe"

_Hydrology and Earth System Sciences, 2018_

## Referee Comment (RC1) · A. Viglione (Referee) · 6 Sep 2018

This paper presents a European data-based analysis of the correlation between a number of atmospheric indices and flood exceedance probabilities at the sub-annual timescale. The novelty of the paper is related to the extent of the study region, i.e., all of Europe. The outcomes are interesting because of the coherent spatial patterns of the identified correlations in climatically different parts of Europe. The paper is well written, properly concise and clear. I believe it can become a very valuable entry for HESS. However, as always, some improvements are possible. My main comments/criticisms/suggestions are the following:

[Figure]

- Title: I do not think that it is possible to easily answer this question, it never is when dealing with extreme value statistics in the real world. Actually, while I consider very interesting the analysis of the correlation between the atmospheric indices and the parameters of flood exceedance probabilities, I am less convinced about the accuracy of flood frequency estimation provided here. The reason is that, in engineering hydrology, I think nobody would fit locally a GEV distribution using a likelihood-based method with no information on its shape parameter. Regional analysis is normally used to improve quantile estimation for high return periods (say 100-years) which is not performed here. I would agree that the paper provides an indication that there is potential for improving flood frequency estimation by including atmospheric dynamics in our models, but I guess there is much more to do to actually improve the existing regional models in use. Maybe this is what the Authors meant but, to me, the title is a bit misleading. In my opinion, a title that focuses on the identified correlations between atmospheric indices and local floods would be better.

- It is always strange, to me, to see studies that use Bayesian inference without using prior information, especially when some very useful prior information is out there. For example, for the GEV shape parameter of the stationary model (but also for the non-stationary one) I recommend using (at least) the "geophysical" prior in Martins and Stedinger (2000). They demonstrate that without a prior on the shape parameter of the GEV, maximum-likelihood estimates (and therefore presumably also Bayesian estimates) for hydrological samples are much less accurate than those obtained with other methods (e.g., L-moments). The "geophysical" prior in Martins and Stedinger (2000) provides "common sense" limits, but even better constraints could be obtained through regional analysis, of course.

- The motivation for assuming that only the location parameter varies in time (through its relationship to the covariates) should be discussed more in detail. Considering the proposed model, with the scale and shape parameters fixed, implies that the variance of the flood series does not change over time (e.g., if the mean annual flood peak increases of 5 m3/s, also the 100-yr flood increases of 5 m3/s, and so all other quantiles). Is this a reasonable assumption? For example, Serago and Vogel (2018) strongly criticize it and propose to use models with the coefficient of variation of the flood series constant in time, since that is consistent with observations in many studies (see the cited literature there). Using a model where CV is constant would be as parsimonious as the one used here and, according to Serago and Vogel (2018), more justified. I suspect that using this other assumption would not invalidate the spatial patterns that are shown in Figures 1 to 4, but would result in very different values in Figures 5 to 7.

- One issue I would also suggest to discuss is the uncertainty in the covariates. The model used here assumes that the covariates are exactly known. If the uncertainty in their knowledge would also be included, would the flood quantile estimates still be more precise than for the classical GEV model?

Additional detailed comments:

Line 20: I would expect that the improvement of estimation of flood probabilities is conditional on how well the covariates can be predicted.

Line 71: I am a little confused by the positive-negative anomalies vs. Northern-southern Europe because the sentence terminates with "during its positive state". Maybe a rephrasing could help.

Line 107: The motivation of using Bayesian inference because of the quantification of uncertainty sounds a bit weak. The quantification of uncertainties is possible also with other methods than Bayesian, which is instead usually selected when subjective preferences or prior information is available (at least by us... statisticians have more profound reasons).

Line 150: The climate covariates are assumed exactly known in the method. Would it be possible to account for the fact that they are stochastic variables as well? I do not ask to change the method but maybe some discussion could be dedicated to this issue

(see main comments).

Line 158: The motivation for assuming that only the location parameter vary in time, i.e., the brevity of records, is not very convincing. The Authors should discuss it more (see main comments).

Line 174: Since Bayesian inference is done here, there is no reason why priors should not be used. For the GEV shape parameter of the stationary model I recommend to use (at least) the "geophysical" prior in Martins and Stedinger (2000) (see main comments).

Line 174: Which non-informative priors are used? Not all of them would result in the same inference. For repeatability, they should be stated.

Line 184: I would also look at the posterior distribution of the slope parameter and do the same as the Authors do here. I would just add a sentence to state that this is not a significance test (which has no meaning in Bayesian statistics).

Line 216: I worry that, for engineering purposes, the estimates of 100-yr floods through GEV without accounting for regional information is not to be recommended.

Line 221: Starting the sentence with "Since a Bayesian framework is used" is confusing because it sounds like saying that uncertainties cannot be quantified with other methods too.

Line 224 (and elsewhere): I would use the wording "posterior mode" instead of "maximum likelihood" because they may not be the same (it depends on the type of non-informative priors that are used). Bayesian posterior predictive distribution of flood peak quantiles or their posterior mean could have also been used. Is there a reason for choosing the posterior mode?

Line 258: Is there any (even speculative) reason for the contradicting patterns for Scandinavia?

Line 265: I also believe that the coherence in space is indicative of a real signal,

however spatial correlation of the flood time-series could be a nuisance here, meaning that one sees the same dynamics in many sites because the same floods are occurring there (and therefore they should count as one site only). Since the spatial patterns are here over very large regions, the spatial correlation of the flood time-series cannot alone be responsible for it. However, I would suggest mentioning the problem.

Line 299: One curiosity. Since the proposed model has constant variance (and the dependence of small and large floods on the covariate is the same in terms of the difference in m3/s) I suspect the relative difference to be affected by catchment area (meaning by the average flow in the river). Is it the case? Of course, since the model is fitted independently to every site, the differences in fitted shape parameters will make this relationship noisier.

Section 3.2: I wonder how much the relative differences calculated here are due to the slope of the regression for the location parameter vs. the estimated shape parameters. Since no priors are used, I suspect that the posterior distributions of the shape parameter can be wide (spanning unreasonable values) and widely different between sites. Maybe a figure/table that also informs the reader about the obtained shape parameters would be useful.

Figure 7: Shouldn't be the classical GEV the same within each column? The credible bounds look different. Have I missed something?

Lines 335: The asymmetry of the credible bounds around the posterior mode is very well expected. If the posterior predictive distribution (or posterior mean) would have been used, that would have lied much more in the center of the bounds.

Line 352: I would add here a brief discussion on the predictability of the covariates since that is needed to make use of the model for prediction.

Line 357: I don't get the meaning of "...leads to highly varying flood quantile estimations for different probabilities of exceedance". Is the sentence referring to variations in time?

Or space? Or between models with different covariates? And, finally, the variability for "different probabilities of exceedance" of flood peaks at one site exists in terms of relative differences. In term of difference in m3/s, there is no variability at all, since in the models only the location parameter can vary. Maybe I just misunderstood. A rephrasing could help.

Line 358: Related to my previous comment on line 299: is it because the catchments in North-West Scandinavia and Britain are smaller than the others? Or is it because of unreasonably large shape parameters of the GEV?

Line 363: It is for me hard to see the decadal-scale variability in Figure 7. Maybe that could be shown in the figure.

Line 402: As an additional challenge (on top of the three that the Authors have listed) I would add the fact that now covariates are assumed perfectly known and should be instead treated as stochastic variables, I think.

References:

Martins, E. S., & Stedinger, J. R. (2000). Generalized maximum‐likelihood generalized extreme‐value quantile estimators for hydrologic data. Water Resources Research, 36(3), 737-744.

Serago, J. M., & Vogel, R. M. (2018). Parsimonious nonstationary flood frequency analysis. Advances in Water Resources, 112, 1-16.

---

## Referee Comment (RC2) · E. Volpi (Referee) · 8 Sep 2018

The manuscript investigates the effectiveness of performing climate-informed extreme value analysis for flood probability estimation at the European scale. More specifically, the Authors analyze the effects of large-scale circulation patterns on seasonal extreme distributions by accounting for the relationship between extreme probabilities and climatic indices. As stated by the Authors, climatic indices are considered in recent literature works to justify or explain a non-stationary behavior depicted by extreme events. In this regard, the innovative contribution of this paper is to perform a large-scale analysis, at a spatial scale that is "comparable" to that of the climatic indices considered

in the work aiming at defining the conditional probability distribution of extreme flood events and proving coherent spatial patterns.

The manuscript is well written and organized; the methodology is almost well described, even if additional details could be included to help for reader understanding, and conclusions are well supported by results. Finally, within the Conclusion Section a detailed list of the limitations of the study is provided. Summarizing, the topic is of interest for the scientific community and the manuscript deserves to be considered for publication in this Journal. I have some comments about the work that are listed in the following paragraph; I hope that they will be helpful for manuscript improvement.

1. The Authors hint in the Introduction Section at the nonstationary framework incorporating climatic indices into flood frequency analysis, but they do not make a clear distinction between periodicity (or cyclo-stationarity) and trends (in the mean or variance). For the sake of clarity, this could be discussed from the very beginning of the manuscript (e.g. at line 49). Are the Authors assuming stationarity which is a "prerequisite to make inference from data", as discussed in detail by the cited papers by Koutsoyiannis and Montanari (2015) and Serinaldi and Kilsby (2015)?

2. At line 136 the Authors define the model driven by climatic indices as "climatic-informed model", justifying this choice based on the fact that "if covariates have a stochastic structure and no deterministic component, the resulting distribution is not truly nonstationary". I do agree on this, as the Authors states at line 135 that the climatic indices are stochastic process not showing clear trends. But, it is expected they are characterized by persistence and/or periodicity. A detailed description of the stochastic behavior of the climatic indices is missing in the manuscript, while they are clearly described from a physical point of view (lines 59-90). E.g. which are the relevant time-period and is the period covered by observations long enough to catch climatic indices periodicities?

[Figure]

3. Even if the aim of the work is to find results at the European scale, I would suggest the Authors to add a figure showing results for a single station, as an illustrative example to explain the methodology and the rationale behind it (e.g. the structure of the climatic informed GEV). Similar to figure 7, it could be of interest to show the evolution in time of the climatic indices (see comments 2) and the performance of classical GEV and climatic-informed GEV, especially for quantile extrapolation, with uncertainty bounds.

4. If I understand correctly, conditional models preferred o classical GEV in Table 1 are those respecting both criteria (minimum value of DIC and significantly different from zero coefficient of linear variation with the climatic indices); this could be highlighted in the result section from the beginning of the section. The number of times (stations) each conditional model is preferred with respect to classical GEV is not so high, being in the best case the $44\%$ and on average at about $20\%$. The use of two criteria does not seem to affect this result much (as in lines 276-280); hence, the evidence of the climatic informed model does not appear to be very strong, even if clear spatial patterns emerge. The latter is the more relevant result, based on my opinion, and this should be stressed in the abstract and conclusion sections.

5. Since spatial patterns are influenced by correlations among climatic indices (that are illustrated in the supplementary material as spatial maps), I suggest the Authors to report in the manuscript a table summarizing cross-correlations among the indices (even if they are not an exhaustive measure of the underling complex physical phenomena).

6. Lines 276-279. DIC is a measure of model evidence; even if the climatic informed model has a smaller value of DIC with respect to classical GEV, the difference among the two values is probably not enough to results in a "strong evidence" of the first model compared to the second one. See, e.g., Kass and Raftery (1995)

where two different interpretations of the Bayes factor are provided.

- Kass, R. E., Raftery, A. E. (1995). Bayes factors. Journal of the american statistical association, 90(430), 773-795.

7. Figure 7 compares conditional (climate informed) and unconditional quantiles considering p=0.01 for three stations. It should be clearly stated that conditional quantiles are computed in this case based on the observed values of the climatic indices year by year.

8. As the climate informed models have a larger number of parameters (one more in this case) to be estimated based on data, it is expected that their uncertainty bounds are larger than those provided by classical GEV. In other words, nonstationarity flood frequency analysis adds an additional component of uncertainty if the model between parameters and covariates is estimated from data and not fully a-priori defined based on additional physical information (Serinaldi and Kilsby, 2015). However, this is not what emerges from figure 7. This issue should be clarified.

9. Lines 329-330. This should be true if the climate indices can be accurately predicted. The issue should be discussed further since it is closely related to the implications of the results presented in the paper for practical applications. Furthermore, I'm asking myself if the improvement in flood quantile estimate at the local scale thanks to climate indices is really significant from a practical point of view given the large uncertainty that characterizes all the estimates (fig. 7); I would like to read a comment on this from the Authors.

10. Line 58. The Authors could also consider the recent paper from Serinaldi et al. (2018) discussing limitations of nonstationary detection based on trend tests.

- Serinaldi, F., Kilsby, C. G., Lombardo, F. (2018). Untenable nonstationarity: An assessment of the fitness for purpose of trend tests in hydrology.

Advances in Water Resources, 111, 132-155.

11. Line 71. A reference is needed.

12. Please definite $t$ after eq. (3).

13. Line 173. Please define what is meant by non-informative priors in this case. If the non-informative prior is a uniform distribution, its support (range of variability of the random parameter) could have effects of posterior distribution and evidence estimation.

14. Eq. (8). $y$ or $Y$?

15. Line 194. Please define $\bar{\theta}$.

16. Line 219. Are the Authors assuming a Gaussian (marginal) distribution for climatic indices? The assumptions on those variables and their stochastic behavior are not clear (see also general comments 2 and 3).

---

## Referee Comment (RC3) · F. Marra (Referee) · 9 Sep 2018

This study presents a methodology to assess and quantify the impact of climatic co-variates in the estimation of time-dependent flood probabilities. The method is tested on a wide sample of catchments in Europe. The paper is clearly and concisely written and the topic is of interest for the readers of this journal.

In my opinion, the study should be seen as an additional step in the efforts of the hydrological community towards a better understanding and quantification of flood risk and, as well underlined by the authors, the spatial consistency of the results indicates some degree of significance in the adopted model. However, further steps are required

before the suggested method can be effectively applied in practice.

Both the reviewers before me pointed out very interesting comments, many of which I happened to share. I come last so I'll try not to overlap.

In general, my main concern derives from the GEV approach that requires a number of hypotheses and, if not integrated within a regionalization framework, is prone to extremely large uncertainties. In addition to the references recommended by Elena Volpi, I may suggest the reading of Marani and Ignaccolo (2015), that provide a different perspective on extreme value analysis and the GEV approach (potentially for nonstationary extremes) that deserves attention.

To conclude, I think the paper definitely deserves publication, but some more discussion and comments on the adopted methods are required. Potentially, some additional analyses could be of help. Below my detailed comments.

- Is the use of climate-informed models contradicting the identical-distribution assumption behind the use of GEV? This perhaps needs to be discussed.

- The inclusion of climate information in the model raises the number of parameters to be estimated to 4. Is there a risk of overfitting?

- How the authors explain that the linear model applied to the scale parameters (rather than location) provides similar results? Shouldn't the two parameters be related one another since the location is related the mean and the scale to the variance of the annual maxima? Is it correct to change one of them and keep the other fixed?

- The GEV approach is highly sensitive to the shape parameter, which is prone to large estimation uncertainty when derived from short data records (50 years), particularly when using the maximum likelihood method. Why not using an L-moments estimation method? Could the inclusion of prior information on the GEV shape parameter improve the accuracy of the results? Perhaps this aspect

should be addressed in the study to check consistency in the significant indices (in the end shape and scale are then used as prior information in the estimation of the climate-informed model parameters).

- A linear model to relate climatic indices and GEV location parameter is chosen. Clearly, more complex models are not recommended due to the limited data sample and overfitting problems, but this represents a simplification of reality. How can this affect the results? This should be discussed.

- What do the authors recommend for situations in which more than one climatic index is significant?

Minor comments:
- Lines 24-28 in the abstract are not easy to read, I suggest to rephrase them;
- Introduction: the proposed method is of interest for (re-)insurance applications and for flood risk management. I think the design applications are not interested since year-by-year variability is not relevant - 169-172: please provide more details for readers not familiar with the technique;

**References**
Marani, M., Ignaccolo, M., 2015. A metastatistical approach to rainfall extremes. Adv. Water Resour. 79, 121–126. https://doi.org/10.1016/j.advwatres.2015.03.001 .

---

## Author Comment (AC1) · 6 Nov 2018

**Reply to Referee #1 Alberto Viglione:**

This paper presents a European data-based analysis of the correlation between a number of atmospheric indices and flood exceedance probabilities at the sub-annual timescale. The novelty of the paper is related to the extent of the study region, i.e., all of Europe. The outcomes are interesting because of the coherent spatial patterns of the identified correlations in climatically different parts of Europe. The paper is well written, properly concise and clear. I believe it can become a very valuable entry for HESS. However, as always, some improvements are possible. My main comments/criticisms/suggestions are the following:

**Response:** We would like to thank Alberto Viglione for his comments. The points he raises are relevant and addressing them will definitely improve our manuscript. We hope that after this discussion (as well as after we revise our manuscript) all issues raised can be clarified.

**General comment 1**
- Title: I do not think that it is possible to easily answer this question, it never is when dealing with extreme value statistics in the real world. Actually, while I consider very interesting the analysis of the correlation between the atmospheric indices and the parameters of flood exceedance probabilities, I am less convinced about the accuracy of flood frequency estimation provided here. The reason is that, in engineering hydrology, I think nobody would fit locally a GEV distribution using a likelihood-based method with no information on its shape parameter. Regional analysis is normally used to improve quantile estimation for high return periods (say 100-years) which is not performed here. I would agree that the paper provides an indication that there is potential for improving flood frequency estimation by including atmospheric dynamics in our models, but I guess there is much more to do to actually improve the existing regional models in use. Maybe this is what the Authors meant but, to me, the title is a bit misleading. In my opinion, a title that focuses on the identified correlations between atmospheric indices and local floods would be better.

**Response:** We found the comment very important and will adopt the reviewer's recommendation concerning the title of our manuscript. We agree that a title focusing on the identified links between circulation indices and local floods better describes our methodology and findings. We will thus change the title accordingly and we will focus on the streamflow-climate interactions. In addition, we will reduce the extrapolation to high return periods and we will calculate streamflow quantiles for a probability of exceedance 0.02 (50-year return period). The common time period of streamflow data and circulation indices is between 50-70 years, so the extrapolation and possible uncertainty from the absence of a regionalization framework is considerably reduced. A comment will be added in the discussion about the possibility of improving quantile estimation by using a regionalization framework. Finally, an informative prior distribution will be used for the shape parameter, in order to constrain the shape parameter from adopting unreasonably high or low values and to improve the GEV fits (see also our reply to general comment 2).

**General comment 2**
- It is always strange, to me, to see studies that use Bayesian inference without using prior information, especially when some very useful prior information is out there. For example, for the GEV shape parameter of the stationary model (but also for the nonstationary one) I recommend using (at least) the "geophysical" prior in Martins and Stedinger (2000). They demonstrate that without a prior on the shape parameter of the GEV, maximum-likelihood estimates (and therefore presumably also Bayesian estimates) for hydrological samples are much less accurate than those obtained with other methods (e.g., L-moments). The "geophysical" prior in Martins and Stedinger (2000) provides "common sense" limits, but even better constraints could be obtained through regional analysis, of course.

**Response:** We found this comment very useful and we repeated the analysis with an informative prior for the shape parameter. The "geophysical" prior of Martins and Stedinger (2000) is bounded in the interval (-0.5, 0.5). There are studies, however, that have found shape parameters of hydro-climatic data higher than 0.5 (e.g. Papalexiou and Koutsoyiannis, 2013). For this reason we considered the restriction of the "geophysical" prior a bit strict and used instead a normal distribution with similar characteristics with those of the "geophysical" prior. The normal distribution allows more extreme shape values but with small probability. The prior we chose is the empirical distribution of the shape parameter found by Papalexiou and Koutsoyiannis (2013) when they fit the GEV to annual precipitation time series worldwide. To our knowledge, it is the study investigating the highest number of hydro-climatic data worldwide. Of course, streamflow may be characterized by slightly higher shape parameter than precipitation. We will comment on this issue in the discussion and conclusions section.

**General comment 3**

- The motivation for assuming that only the location parameter varies in time (through its relationship to the covariates) should be discussed more in detail. Considering the proposed model, with the scale and shape parameters fixed, implies that the variance of the flood series does not change over time (e.g., if the mean annual flood peak increases of 5 m3/s, also the 100-yr flood increases of 5 m3/s, and so all other quantiles).
Is this a reasonable assumption? For example, Serago and Vogel (2018) strongly criticize it and propose to use models with the coefficient of variation of the flood series constant in time, since that is consistent with observations in many studies (see the cited literature there). Using a model where CV is constant would be as parsimonious as the one used here and, according to Serago and Vogel (2018), more justified. I suspect that using this other assumption would not invalidate the spatial patterns that are shown in Figures 1 to 4, but would result in very different values in Figures 5 to 7.

**Response:** The reviewer is right, indeed both the location and scale parameter are expected to change based on the climate state. However, we tested for significance of varying scale parameter by running the model with both location and scale variable. This preliminary study showed only very few cases with significant slopes of the scale parameter. For this reason and for reasons of parsimony, we decided to keep the scale parameter stable and to condition only the location parameter on the climate indices.
The model with constant coefficient of variation (CV) is an interesting alternative to the model that we present in our manuscript. However, investigating additionally this model would lead to a different and considerably extended manuscript. We feel that such a change is beyond the scope of this paper. We will comment on the possibility of a constant coefficient of variation (CV) in the discussion and conclusions section.

**General comment 4**

- One issue I would also suggest to discuss is the uncertainty in the covariates. The model used here assumes that the covariates are exactly known. If the uncertainty in their knowledge would also be included, would the flood quantile estimates still be more precise than for the classical GEV model?

**Response:** Thank you for this comment. In our manuscript we investigate only contemporaneous relationships between climate indices and flood peaks and do not focus on prediction. Our goal is mainly to identify spatial patterns of these relationships. For this reason we assume that covariates are exactly known. Of course if one wants to use the current model in a predictive mode, the uncertainty in the covariates must be additionally considered. We will add a comment on this issue in the discussion.

Additional detailed comments:
Line 20: I would expect that the improvement of estimation of flood probabilities is conditional on how well the covariates can be predicted.

**Response:** The reviewer is right. We will rephrase this sentence in analogy to our detailed response to general comment 4.

Line 71: I am a little confused by the positive-negative anomalies vs. Northernsouthern Europe because the sentence terminates with "during its positive state".
Maybe a rephrasing could help.

**Response:** In the new version a change will be made from "Particularly NAO has been shown to significantly influence the European winter climate with positive (negative) anomalies of moisture fluxes, cyclone passages and precipitation over northern (southern) Europe during its positive state" to "Particularly NAO has been shown to significantly influence the European winter climate: its positive state has been linked to positive (negative) anomalies of moisture fluxes, cyclone passages and precipitation over northern (southern) Europe".

Line 107: The motivation of using Bayesian inference because of the quantification of uncertainty sounds a bit weak. The quantification of uncertainties is possible also with other methods than Bayesian, which is instead usually selected when subjective preferences or prior information is available (at least by us... statisticians have more profound reasons).

**Response:** We will add a sentence stating that the Bayesian framework is justified since prior information on the shape parameter exists in literature.

Line 150: The climate covariates are assumed exactly known in the method. Would it be possible to account for the fact that they are stochastic variables as well? I do not ask to change the method but maybe some discussion could be dedicated to this issue (see main comments).

**Response:** See our reply to general comment 4.

Line 158: The motivation for assuming that only the location parameter vary in time, i.e., the brevity of records, is not very convincing. The Authors should discuss it more (see main comments).

**Response:** See our reply to general comment 3.

Line 174: Since Bayesian inference is done here, there is no reason why priors should not be used. For the GEV shape parameter of the stationary model I recommend to use (at least) the "geophysical" prior in Martins and Stedinger (2000) (see main comments).

**Response:** See our reply to general comment 2.

Line 174: Which non-informative priors are used? Not all of them would result in the same inference. For repeatability, they should be stated.

**Response:** We will add a description of the prior distributions used: uniform priors for the location and scale parameters and a normal informative prior for the shape parameter (see also our reply to general comment 2.

Line 184: I would also look at the posterior distribution of the slope parameter and do the same as the Authors do here. I would just add a sentence to state that this is not a significance test (which has no meaning in Bayesian statistics).

**Response:** We will add a comment on the fact that this is not a significance test.

Line 216: I worry that, for engineering purposes, the estimates of 100-yr floods through GEV without accounting for regional information is not to be recommended.

**Response:** See our reply to general comment 1.

Line 221: Starting the sentence with "Since a Bayesian framework is used" is confusing because it sounds like saying that uncertainties cannot be quantified with other methods too.

**Response:** We will rephrase this sentence.

Line 224 (and elsewhere): I would use the wording "posterior mode" instead of "maximum likelihood" because they may not be the same (it depends on the type of noninformative priors that are used). Bayesian posterior predictive distribution of flood peak quantiles or their posterior mean could have also been used. Is there a reason for choosing the posterior mode?

**Response:** The posterior mode was used in order to make results comparable with those of frequentist approaches. In the revised manuscript we will use the posterior median of flood peak quantiles, because it is more representative of the posterior distribution.

Line 258: Is there any (even speculative) reason for the contradicting patterns for Scandinavia?

**Response:** We will shortly discuss potential reasons for the deviant behavior in the revised manuscript version. In this regard, we will particularly refer to the special catchment characteristics in Scandinavia.
The fact, that the integration of seasonal mean climate indices in the flood frequency analysis improves the extreme value distributions for most catchments in Central Europe indicates, that catchment wetness (due to variations of seasonal precipitation sums) might play an important role for flood generation in those regions. In contrast, Scandinavian rivers usually have small catchments and are particularly fed by snow melt in spring and both, temperature and precipitation, may be important for runoff generation. A positive state of SCA is associated with negative precipitation anomalies (Supplement 2), but also with positive anomalies of temperature and incoming solar radiation (not shown) and vice versa. Thus intense snow melt events might be more likely during dry conditions associated with a positive SCA index.

Line 265: I also believe that the coherence in space is indicative of a real signal, however spatial correlation of the flood time-series could be a nuisance here, meaning that one sees the same dynamics in many sites because the same floods are occurring there (and therefore they should count as one site only). Since the spatial patterns are here over very large regions, the spatial correlation of the flood time-series cannot alone be responsible for it. However, I would suggest mentioning the problem.

**Response:** We agree that spatial correlation of floods plays a role for the detected coherence particularly for smaller regions, i.e. nearby gauges. We will add a comment on the spatial correlation of the flood time-series.

Line 299: One curiosity. Since the proposed model has constant variance (and the dependence of small and large floods on the covariate is the same in terms of the difference in m3/s) I suspect the relative difference to be affected by catchment area (meaning by the average flow in the river). Is it the case? Of course, since the model is fitted independently to every site, the differences in fitted shape parameters will make this relationship noisier.

**Response:** Since the difference between streamflow quantiles for high and medium covariate is normalized by the streamflow quantile for medium covariate we were not

expecting that the catchment size plays a role in the percent relative differences. We assume that the high relative differences are due to a stronger influence of the climatic indices.

Section 3.2: I wonder how much the relative differences calculated here are due to the slope of the regression for the location parameter vs. the estimated shape parameters. Since no priors are used, I suspect that the posterior distributions of the shape parameter can be wide (spanning unreasonable values) and widely different between sites. Maybe a figure/table that also informs the reader about the obtained shape parameters would be useful.

**Response:** We will add a table with summary statistics of the shape parameter.

Figure 7: Shouldn't be the classical GEV the same within each column? The credible bounds look different. Have I missed something?

**Response:** We are sorry, this was a typo error noticed by the reviewer and was corrected.

Lines 335: The asymmetry of the credible bounds around the posterior mode is very well expected. If the posterior predictive distribution (or posterior mean) would have been used, that would have lied much more in the center of the bounds.

**Response:** The posterior median is used now. Indeed credible bounds are less asymmetrical.

Line 352: I would add here a brief discussion on the predictability of the covariates since that is needed to make use of the model for prediction.

**Response:** See our reply to general comment 4.

Line 357: I don't get the meaning of "...leads to highly varying flood quantile estimations for different probabilities of exceedance". Is the sentence referring to variations in time? Or space? Or between models with different covariates? And, finally, the variability for "different probabilities of exceedance" of flood peaks at one site exists in terms of relative differences. In term of difference in m3/s, there is no variability at all, since in the models only the location parameter can vary. Maybe I just misunderstood. A rephrasing could help.

**Response:** Indeed the variability concerns the relative differences. The sentence is referring to the comparison between the classical GEV and the climate-informed models. We will rephrase it to make this clearer.

Line 358: Related to my previous comment on line 299: is it because the catchments in North-West Scandinavia and Britain are smaller than the others? Or is it because of unreasonably large shape parameters of the GEV?

**Response:** The highly varying results in this area are in our opinion the result of a more important influence of the circulation indices. No influence of the catchment size was found.

Line 363: It is for me hard to see the decadal-scale variability in Figure 7. Maybe that could be shown in the figure.

**Response:** In the revised manuscript we will add a figure with the evolution in time of the climate indices, showing also their decadal-scale variability. We think that this will help the readers better interpret figure 7.

Line 402: As an additional challenge (on top of the three that the Authors have listed) I would add the fact that now covariates are assumed perfectly known and should be instead treated as stochastic variables, I think.

**Response:** See our reply to general comment 4.

**References:**

Papalexiou, S. M. and Koutsoyiannis, D.: Battle of extreme value distributions: A global survey on extreme daily rainfall, Water Resour. Res., 49(1), 187–201, doi:10.1029/2012WR012557, 2013.

---

## Author Comment (AC2) · 6 Nov 2018

**Reply to Referee #2 Elena Volpi:**

The manuscript investigates the effectiveness of performing climate-informed extreme value analysis for flood probability estimation at the European scale. More specifically, the Authors analyze the effects of large-scale circulation patterns on seasonal extreme distributions by accounting for the relationship between extreme probabilities and climatic indices. As stated by the Authors, climatic indices are considered in recent literature works to justify or explain a non-stationary behavior depicted by extreme events.
In this regard, the innovative contribution of this paper is to perform a large-scale analysis, at a spatial scale that is "comparable" to that of the climatic indices considered in the work aiming at defining the conditional probability distribution of extreme flood events and proving coherent spatial patterns.
The manuscript is well written and organized; the methodology is almost well described, even if additional details could be included to help for reader understanding, and conclusions are well supported by results. Finally, within the Conclusion Section a detailed list of the limitations of the study is provided. Summarizing, the topic is of interest for the scientific community and the manuscript deserves to be considered for publication in this Journal. I have some comments about the work that are listed in the following paragraph; I hope that they will be helpful for manuscript improvement.

**Response:** We would like to thank Elena Volpi for her comments. In the revised manuscript we will follow most of the reviewer's recommendations, since this will definitely improve our study. Below, we provide justification for some suggestions that we do not follow. We see from the comments of the reviewer that some parts of our study need a more detailed explanation. In our revised manuscript we will provide these additional details.

**General comment 1**
1. The Authors hint in the Introduction Section at the nonstationary framework incorporating climatic indices into flood frequency analysis, but they do not make a clear distinction between periodicity (or cyclo-stationarity) and trends (in the mean or variance). For the sake of clarity, this could be discussed from the very beginning of the manuscript (e.g. at line 49). Are the Authors assuming stationarity which is a "prerequisite to make inference from data", as discussed in detail by the cited papers by Koutsoyiannis and Montanari (2015) and Serinaldi and Kilsby (2015)?
**General comment 2**
2. At line 136 the Authors define the model driven by climatic indices as "climatic informed model", justifying this choice based on the fact that "if covariates have a stochastic structure and no deterministic component, the resulting distribution is not truly nonstationary". I do agree on this, as the Authors states at line 135 that the climatic indices are stochastic process not showing clear trends. But, it is expected they are characterized by persistence and/or periodicity. A detailed description of the stochastic behavior of the climatic indices is missing in the manuscript, while they are clearly described from a physical point of view (lines 59-90). E.g. which are the relevant time-period and is the period covered by observations long enough to catch climatic indices periodicities?

**Response to general comments 1 and 2:**
We want to thank the reviewer for these interesting comments. In our manuscript we acknowledge the issue that models conditional on time-varying covariates with a stochastic structure can be stationary, even if the probability density function changes in consequent years. However, we feel that addressing the issue of stationarity/nonstationarity (and thus ergodicity) and the stochastic structure of the covariates in adequate detail would change considerably the focus of our manuscript and we prefer not to make this addition. For this reason we chose the term "classical" and "climate-informed models" and we do not refer to stationary/nonstationary models. We will consider going more in this direction in our future work.

In the revised manuscript we will add a more detailed comment stating that "a detailed description of the stochastic behaviour of the circulation indices would be needed in order to argue in the direction of stationarity or nonstationarity". Furthermore, we will add a comment in the introduction in order to make a more clear distinction between periodicity and trends in the description on nonstationary models. Finally, we will add a figure illustrating the evolution in time of seasonal climate indices.

**General comment 3**
3. Even if the aim of the work is to find results at the European scale, I would suggest the Authors to add a figure showing results for a single station, as an illustrative example to explain the methodology and the rationale behind it (e.g. the structure of the climatic informed GEV). Similar to figure 7, it could be of interest to show the evolution in time of the climatic indices (see comments 2) and the performance of classical GEV and climatic-informed GEV, especially for quantile extrapolation, with uncertainty bounds.

**Response:** We will adopt the recommendation of the reviewer and we will add an example figure illustrating the performance of classical GEV and climatic-informed GEV with uncertainty bounds for possible covariate values (additionally to the time-varying uncertainty bounds shown in figure 7). A figure with the evolution in time of the climate indices will also be added (see also our response to general comments 1-2). However, we do not plan to add a figure explaining the climate-informed model. As we also state in the introduction, during the last years there have been many studies applying such a conditional framework to single or a few stations. We feel there is enough published material explaining this methodology.

4. If I understand correctly, conditional models preferred o classical GEV in Table 1 are those respecting both criteria (minimum value of DIC and significantly different from zero coefficient of linear variation with the climatic indices); this could be highlighted in the result section from the beginning of the section. The number of times (stations) each conditional model is preferred with respect to classical GEV is not so high, being in the best case the 44% and on average at about 20%. The use of two criteria does not seem to affect this result much (as in lines 276-280); hence, the evidence of the climatic informed model does not appear to be very strong, even if clear spatial patterns emerge. The latter is the more relevant result, based on my opinion, and this should be stressed in the abstract and conclusion sections.

**Response:** We will adopt this suggestion and in the revised manuscript we will highlight the selection criteria in the result section. We will stress more clearly that the effect of each index independently is not always high (in many cases it affects a 20% of the database). However, the number of stations affected by at least one index significantly is much higher, especially in winter. We feel that this is a result that indicates a real influence of the circulation indices to the streamflow extremes.

5. Since spatial patterns are influenced by correlations among climatic indices (that are illustrated in the supplementary material as spatial maps), I suggest the Authors to report in the manuscript a table summarizing cross-correlations among the indices (even if they are not an exhaustive measure of the underling complex physical phenomena).

**Response:** We will adopt this suggestion and we will add a Table in the Supplementary material summarising linear correlations between the seasonal indices.

6. Lines 276-279. DIC is a measure of model evidence; even if the climatic informed model has a smaller value of DIC with respect to classical GEV, the difference among the two values is probably not enough to results in a "strong evidence" of the first model compared to the second one. See, e.g., Kass and Raftery (1995) where two different interpretations of the Bayes factor are provided.
• Kass, R. E., Raftery, A. E. (1995). Bayes factors. Journal of the american statistical association, 90(430), 773-795.

**Response:** Since we are using two criteria, the DIC and slope significance, we feel that the evidence is enough in the case of pairwise comparison with the classical model. In the case of comparison between the climate-informed models (for example Fig. 3-4) a comment will be added highlighting that more evidence may be required for a formal decision. In the discussion and conclusions section we already mention some of the limitations of DIC. We will discuss further the issue of "strong evidence" for model selection.

7. Figure 7 compares conditional (climate informed) and unconditional quantiles considering p=0.01 for three stations. It should be clearly stated that conditional quantiles are computed in this case based on the observed values of the climatic indices year by year.

**Response:** We will add a comment stating this year by year quantile calculation.

8. As the climate informed models have a larger number of parameters (one more in this case) to be estimated based on data, it is expected that their uncertainty bounds are larger than those provided by classical GEV. In other words, nonstationarity flood frequency analysis adds an additional component of uncertainty if the model between parameters and covariates is estimated from data and not fully a-priori defined based on additional physical information (Serinaldi and Kilsby, 2015). However, this is not what emerges from figure 7. This issue should be clarified.

**Response:** This was a very helpful comment. We realise that figure 7 and the conditional/unconditional uncertainty bounds need to be further discussed. Our figure does not contradict the findings and discussion of Serinaldi and Kilsby (2015).
The range of the uncertainty bounds is an interplay between the model complexity and the additional information provided by the more complex models. The relation between the two is not always trivial. In general, more complex models not providing extra information are expected to lead to an increase in uncertainty. More complex models providing "adequate" additional information are expected to lead to decreased uncertainty.
In figure 7, uncertainty bounds are narrower in the case of the "best" conditional models (e.g. plot A1). The bounds are also narrower for common values of the climate indices, while they can be larger than the classical case when extrapolations are made to uncommon (high and low) index values. In this case not enough extra information is available and uncertainty increases.
We will discuss this issue in more detail in the manuscript. We will furthermore add one more explanatory figure showing the uncertainty bounds versus the covariate values (see also our reply to general comment 3). This will better clarify our point and findings.

9. Lines 329-330. This should be true if the climate indices can be accurately predicted. The issue should be discussed further since it is closely related to the implications of the results presented in the paper for practical applications. Furthermore, I'm asking myself if the improvement in flood quantile estimate at the local scale thanks to climate indices is really significant from a practical point of view given the large uncertainty that characterizes all the estimates (fig. 7); I would like to read a comment on this from the Authors.

**Response:** A comment will be added in the discussion explaining the effect of the index uncertainty in the limitations of the study. Indeed if one wants to predict the indices for the next season in order to use them for the estimation of streamflow quantiles, uncertainties will be higher.

10. Line 58. The Authors could also consider the recent paper from Serinaldi et al. (2018) discussing limitations of nonstationary detection based on trend tests.
• Serinaldi, F., Kilsby, C. G., Lombardo, F. (2018). Untenable nonstationarity: An assessment of the fitness for purpose of trend tests in hydrology.

**Response:** Thanks for this very interesting paper. We will consider it for the discussion of conditional / nonstationary model.

11. Line 71. A reference is needed.

**Response:** We will add it.

12. Please definite t after eq. (3).

**Response:** We will correct that.

13. Line 173. Please define what is meant by non-informative priors in this case. If the non-informative prior is a uniform distribution, its support (range of variability of the random parameter) could have effects of posterior distribution and evidence estimation.

**Response:** We will add a description of the prior distributions. In the revised manuscript we will use uniform priors for the location and scale parameters and a normal informative prior for the shape parameter.

14. Eq. (8). y or Y ?

**Response:** We will replace y with Y.

15. Line 194. Please define $\bar{\bar{\theta}}$.

**Response:** We will add this definition.

16. Line 219. Are the Authors assuming a Gaussian (marginal) distribution for climatic indices? The assumptions on those variables and their stochastic behaviour are not clear (see also general comments 2 and 3).

**Response:** We are currently not making an assumption about the marginal distribution of climate indices. We will add a figure, however, in the supplementary material with the histograms of all seasonal indices so that the readers can get a better idea about their distribution.

---

## Author Comment (AC3) · 6 Nov 2018

**Reply to Referee #3 Francesco Marra:**

This study presents a methodology to assess and quantify the impact of climatic covariates in the estimation of time-dependent flood probabilities. The method is tested on a wide sample of catchments in Europe. The paper is clearly and concisely written and the topic is of interest for the readers of this journal.
In my opinion, the study should be seen as an additional step in the efforts of the hydrological community towards a better understanding and quantification of flood risk and, as well underlined by the authors, the spatial consistency of the results indicates some degree of significance in the adopted model. However, further steps are required before the suggested method can be effectively applied in practice. Both the reviewers before me pointed out very interesting comments, many of which I happened to share. I come last so I'll try not to overlap.

**Response:** We would like to thank Francesco Marra for his comments. The referee has shared very valid concerns that we hope to address in a revised version of our manuscript.

**General comment 1**
In general, my main concern derives from the GEV approach that requires a number of hypotheses and, if not integrated within a regionalization framework, is prone to extremely large uncertainties.

**Response:** The concern of the reviewer is valid. Indeed, a regional framework is commonly used in order to extrapolate inference to higher return periods. Here, focusing more on identifying coherent patterns in space, we use a local framework, which is able to recognise significant influence of certain indices to the extreme streamflow quantiles in certain regions of Europe. In order to reduce uncertainty, we will constrain our analysis to the 50-year return period which covers the data length: in our study the overlapping period between climate indices and streamflow time series is between 50 and 70 years. Furthermore, in order to improve GEV fits in the revised manuscript, prior information for the shape parameter will be included in the analysis (see also our response to main comment 2 of Alberto Viglione).

In addition to the references recommended by Elena Volpi, I may suggest the reading of Marani and Ignaccolo (2015), that provide a different perspective on extreme value analysis and the GEV approach (potentially for nonstationary extremes) that deserves attention.

**Response:** Thanks for this very interesting paper. We will consider it for the discussion of alternative models.

To conclude, I think the paper definitely deserves publication, but some more discussion and comments on the adopted methods are required. Potentially, some additional analyses could be of help. Below my detailed comments.

• Is the use of climate-informed models contradicting the identical-distribution assumption behind the use of GEV? This perhaps needs to be discussed.

**Response:** The condition of independent and identically distributed observations of GEV can be relaxed to include parameters conditioned on time-varying covariates. This can be achieved by converting the original data to generalized "residuals" that are identically distributed (Katz et al., 2002). We will add a comment on this issue in the revised manuscript.

• The inclusion of climate information in the model raises the number of parameters to be estimated to 4. Is there a risk of overfitting?

**Response:** Our climate-informed models have one more parameter compared to the classical GEV, derived by physical reasoning (the climate indices influence the climate and

hydrology of the European area). According to e.g. Katz et al. (2002), such models are reasonable. Furthermore, our conditional models are compared to the classical GEV by means of the DIC, which penalizes model complexity. We are thus confident that the possibility of overfitting is minimized.

• How the authors explain that the linear model applied to the scale parameters (rather than location) provides similar results? Shouldn't the two parameters be related one another since the location is related the mean and the scale to the variance of the annual maxima? Is it correct to change one of them and keep the other fixed?

**Response:** The reviewer is right, indeed both the location and scale parameter are expected to change based on the climate state. However, we tested for significance of varying scale parameter by running the model with both location and scale variable. This preliminary study showed only very few cases with significant slopes of the scale parameter. For this reason and for reasons of parsimony, we decided to keep the scale parameter stable and to condition only the location parameter on the climate indices.

• The GEV approach is highly sensitive to the shape parameter, which is prone to large estimation uncertainty when derived from short data records (50 years), particularly when using the maximum likelihood method. Why not using an Lmoments estimation method? Could the inclusion of prior information on the GEV shape parameter improve the accuracy of the results? Perhaps this aspect should be addressed in the study to check consistency in the significant indices (in the end shape and scale are then used as prior information in the estimation of the climate-informed model parameters).

**Response:** Thanks for this interesting comment. The Bayesian approach was chosen because of its advantages concerning quantification of uncertainty. The choice of a likelihood-based method offers additionally a straightforward way of including covariates in the frequency analysis. The reviewer is right, inclusion of prior information on the GEV shape parameter improves the accuracy of the results. In our revised manuscript we will use an informative prior for our classical and climate-informed GEV. Please see our reply to general comment 2 of Alberto Viglione for more details on the prior distribution chosen. One more detail that we would like to highlight is that the posterior distributions of the parameters of the classical GEV are not currently used as priors for the climate-informed case. Each model is fitted independently.

• A linear model to relate climatic indices and GEV location parameter is chosen. Clearly, more complex models are not recommended due to the limited data sample and overfitting problems, but this represents a simplification of reality. How can this affect the results? This should be discussed.

**Response:** Of course, a linear univariate model is always a simplification and non-linear relationships between the state of climate indices and the European surface climate might exist, which also result in a non-linear influence on streamflow extremes. The choice of a linear model can affect results, since the effect of the covariate on the dependent variable may be overestimated or underestimated for certain covariate values. However, linear methods are frequently applied, particularly if the data sets do not allow to fit more complex models.
In our discussion and conclusions section we mention the possibility of an asymmetrical model for the positive and negative phase of the indices. In Lines 389-392 we write:
"A symmetrical influence of the positive and negative phases of the climate indices on the extreme value distribution has been assumed in this study. However, an asymmetrical relation may better describe the effect of certain climate modes on streamflow extremes. For example, Sun et al. (2014) used an asymmetric piecewise-linear regression to account for the different effects of El Niño and La Niña on rainfall extremes in Southeast Queensland, Australia".

In our revised manuscript we will extend this discussion to point out more clearly the limitations because of choosing a linear model as discussed above.

• What do the authors recommend for situations in which more than one climatic index is significant?

**Response:** A multi-linear model for such cases is also possible. For example, in our study there seems to be a significant influence of both NAO and SCA in winter in Central Europe. We comment on this issue in our discussion (Lines 383-387):
"Single covariate models were developed, focusing on the separate effect of each individual climate mode. The methodology can be extended to a model considering several covariates at the same time. In that case, dependencies between the covariates, if existent, should be taken into consideration. López and Francés (2013) overcame this problem by using the principal components of climatic indices as covariates for the flood frequency analysis".

Minor comments:
- Lines 24-28 in the abstract are not easy to read, I suggest to rephrase them;

**Response:** We will rephrase them according to the revised manuscript.

- Introduction: the proposed method is of interest for (re-)insurance applications and for flood risk management. I think the design applications are not interested since year-by-year variability is not relevant

**Response:** Indeed the proposed framework makes more sense when the year-to-year variability is investigated. We will omit "engineering design" in line 51.

- 169-172: please provide more details for readers not familiar with the technique;

**Response:** We will add more details on the No-U-Turn Sampler-Hamiltonian Monte Carlo approach.

**References:**
Katz, R. W., Parlange. M. B. and Naveau, P.: Statistics of extremes in hydrology. Advances in Water Resources 25(8-12): 1287 – 1304, 2002.

---

## Author Response (AR1)

**List of all relevant changes**

The following major changes have been made to the manuscript:
(R1, R2 and R3 indicate suggestions by reviewer 1, 2 and 3, respectively).

1. The title of the manuscript was changed (R1).
2. All computations were made from the beginning using an informative prior for the shape parameter (R1, R3). Fig. 1-8, S7-S10 and Tables 1, 2, 6 were updated. No major changes were observed in spatial patterns. Shape parameters were constrained to reasonable limits.
3. Table 4 with summary statistics of the shape parameter was added (R1).
4. The median flood quantiles were used as point estimate instead of the modal estimate used before (R1). Figures 5, 6, 7 and Table 6 were updated. Results are similar to those in the initial manuscript but uncertainty bounds in Fig. 7 are less skewed.
5. A probability of exceedance 0.02 instead of 0.01 was used for the calculation of flood quantiles (R1, R3). Figures 5, 6, 7 and Table 6 were updated. Spatial patterns are similar to those in the initial manuscript. The absolute values of percent level differences in flood quantiles were decreased in comparison to a probability 0.01.
6. Table S1 was added in the supplementary material, showing correlations between seasonal climate indices (R2).
7. Fig. 8, comparing streamflow quantiles for the classical and a conditional model for a single station, was added (R2).
8. Fig. S5 was added in the supplementary material. This figure shows the evolution in time of the climate indices (R2), along with their decadal-scale variability (R1).
9. Fig. S6 showing histograms of the seasonal climate indices was added in the supplementary material (R2).
10. Extensive changes in discussion and conclusions were made (R1, R2, and R3).

In addition to the above major changes, numerous smaller changes have been made and are shown in the marked manuscript (R1, R2, and R3).

[revised manuscript text omitted]

**Figure S5S7: Same as Fig. 1 but for the monthly indices at the month of the season maximum streamflow.**

[Figure]

| summer | autumn |
| --- | --- |

NAO

EA

EA/WR

SCA

POL

● Nonsignificant (no DIC)  ▲ Positive slope (both criteria)

● Nonsignificant (only DIC)  ▼ Negative slope (both criteria)

[Figure]

**Figure S8: Same as Fig.  S7 but for the summer and autumn season.**

[Figure]

[Figure]

**Figure S7S9: Same as Fig. 3 but for the monthly indices at the month of the season maximum streamflow.**

[Figure]

[Figure]

**Figure S10: Same as Fig. 4 but for the monthly indices at the month of the season maximum streamflow.**

**Table S1: Pearson correlation coefficient between the indices examined. Statistically significant results at the 5% level are highlighted.**

| Indices | Winter | Spring | Summer | Autumn |
|---|---|---|---|---|
| **NAO - EA** | 0.126 | 0.026 | -0.336* | -0.207 |
| **NAO – EA/WR** | 0.085 | 0.176 | 0.252* | 0.155 |
| **NAO - SCA** | -0.314* | 0.065 | 0.253* | 0.022 |

| | | | | |
|---|---|---|---|---|
| **NAO - POL** | -0.235 | -0.029 | -0.069 | 0.144 |
| **EA – EA/WR** | -0.028 | -0.003 | -0.446* | -0.095 |
| **EA – SCA** | 0.023 | -0.166 | -0.343* | -0.040 |

**References for Supplementary Material**

Harris, I., Jones, P. D., Osborn, T. J. and Lister, D. H.: Updated high-resolution grids of monthly climatic observations - the CRU TS3.10 Dataset, Int. J. Climatol., 34(3), 623–642, doi:10.1002/joc.3711, 2014.

Kalnay, E., Kanamitsu, M., Kistler, R., Collins, W., Deaven, D., Gandin, L., Iredell, M., Saha, S., White, G., Woollen, J., Zhu, Y., Leetmaa, A., Reynolds, R., Chelliah, M., Ebisuzaki, W., Higgins, W., Janowiak, J., Mo, K. C., Ropelewski, C., Wang, J., Jenne, R. and Joseph, D.: The NCEP/NCAR 40-Year Reanalysis Project, Bull. Am. Meteorol. Soc., 77(3), 437–471, doi:10.1175/1520-0477(1996)077<0437:TNYRP>2.0.CO;2, 1996.

Steirou, E., Gerlitz, L., Apel, H. and Merz, B.: Links between large-scale circulation patterns and streamflow in Central Europe: A review, J. Hydrol., 549, doi:10.1016/j.jhydrol.2017.04.003, 2017.

**Reply to Referee #1 Alberto Viglione:**

This paper presents a European data-based analysis of the correlation between a number of atmospheric indices and flood exceedance probabilities at the sub-annual timescale. The novelty of the paper is related to the extent of the study region, i.e., all of Europe. The outcomes are interesting because of the coherent spatial patterns of the identified correlations in climatically different parts of Europe. The paper is well written, properly concise and clear. I believe it can become a very valuable entry for HESS. However, as always, some improvements are possible. My main comments/criticisms/suggestions are the following:

**Response:** We would like to thank Alberto Viglione for his comments. The points he raised are constructive and addressing them definitely improved our manuscript.

**General comment 1**
- Title: I do not think that it is possible to easily answer this question, it never is when dealing with extreme value statistics in the real world. Actually, while I consider very interesting the analysis of the correlation between the atmospheric indices and the parameters of flood exceedance probabilities, I am less convinced about the accuracy of flood frequency estimation provided here. The reason is that, in engineering hydrology, I think nobody would fit locally a GEV distribution using a likelihood-based method with no information on its shape parameter. Regional analysis is normally used to improve quantile estimation for high return periods (say 100-years) which is not performed here. I would agree that the paper provides an indication that there is potential for improving flood frequency estimation by including atmospheric dynamics in our models, but I guess there is much more to do to actually improve the existing regional models in use. Maybe this is what the Authors meant but, to me, the title is a bit misleading. In my opinion, a title that focuses on the identified correlations between atmospheric indices and local floods would be better.

**Response:** We found the comment very important and we adopted the reviewer's recommendation concerning the title of our manuscript. The title was changed to "Climate influences on flood probabilities across Europe" that focuses on the spatial aspect of our analysis and on streamflow-climate interactions. In addition, we reduced the extrapolation towards high return periods and we calculated streamflow quantiles for a probability of exceedance 0.02 (50-year return period). The common time period of streamflow data and circulation indices is between 50-70 years, so the extrapolation and possible uncertainty from the absence of a regionalization framework is considerably reduced. Furthermore, an informative prior distribution was used for the shape parameter, in order to constrain the shape parameter from adopting unreasonably high or low values and to improve the GEV fits (see also our reply to general comment 2).
Finally, a comment was added in the discussion about the possibility of improving quantile estimation by using a regionalization framework (Lines 384-389 of the revised manuscript): "In this study, a local, site-specific flood frequency model was developed. This model allowed to identify spatial coherence in relations between streamflow extremes and large-scale atmospheric patterns. However, a shortcoming of this methodology is the high uncertainty of streamflow estimates for high probabilities of exceedance (corresponding for example to the 100- or 200-year flood). Instead of a local framework, a regional framework can be alternatively implemented. The latter, by considering all available streamflow information in a region, decreases uncertainty and offers the possibility of improving streamflow quantile estimation".

**General comment 2**
- It is always strange, to me, to see studies that use Bayesian inference without using prior information, especially when some very useful prior information is out there. For example, for the GEV shape parameter of the stationary model (but also for the nonstationary one) I

recommend using (at least) the "geophysical" prior in Martins and Stedinger (2000). They demonstrate that without a prior on the shape parameter of the GEV, maximum-likelihood estimates (and therefore presumably also Bayesian estimates) for hydrological samples are much less accurate than those obtained with other methods (e.g., L-moments). The "geophysical" prior in Martins and Stedinger (2000) provides "common sense" limits, but even better constraints could be obtained through regional analysis, of course.

**Response:** We found this comment very useful and we repeated the analysis with an informative prior for the shape parameter. The following paragraph was added in the "Flood frequency analysis – Competing models" section (lines 169-175):
"For the shape parameter an informative normal distribution with mean 0.093 and standard deviation 0.12 is used. This distribution is adopted from a global study of extreme rainfall by Papalexiou and Koutsoyiannis (2013), which, to our knowledge, summarizes an analysis of shape parameters using the largest number of stations with hydrological data worldwide. Although rainfall extremes may be characterized by slightly different shape parameter than those of streamflow, our informative prior is very close to the "geophysical prior" of Martins and Stedinger (2000), which is often used to restrict the range of shape parameters based on previous hydrological experience (Renard et al. 2013). The latter prior was not preferred because it is bounded to the interval (-0.5, 0.5), while the distribution of Papalexiou and Koutsoyiannis (2013) allows more extreme shape values with a low probability".

**General comment 3**
- The motivation for assuming that only the location parameter varies in time (through its relationship to the covariates) should be discussed more in detail. Considering the proposed model, with the scale and shape parameters fixed, implies that the variance of the flood series does not change over time (e.g., if the mean annual flood peak increases of 5 m3/s, also the 100-yr flood increases of 5 m3/s, and so all other quantiles).
Is this a reasonable assumption? For example, Serago and Vogel (2018) strongly criticize it and propose to use models with the coefficient of variation of the flood series constant in time, since that is consistent with observations in many studies (see the cited literature there). Using a model where CV is constant would be as parsimonious as the one used here and, according to Serago and Vogel (2018), more justified. I suspect that using this other assumption would not invalidate the spatial patterns that are shown in Figures 1 to 4, but would result in very different values in Figures 5 to 7.

**Response:** The reviewer is right, indeed both the location and scale parameter are expected to change based on the climate state. However, we tested for significance of varying scale parameter by running the model with both location and scale variable. This preliminary study showed only very few cases with significant slopes of the scale parameter. For this reason and for reasons of parsimony, we decided to keep the scale parameter stable and to condition only the location parameter on the climate indices. A comment on this issue was added in the "Flood frequency analysis – Competing models" section (lines 149-152):
"A preliminary analysis considering the effect of a covariate on both the location and scale parameter (cf. section 2.3 below) did not provide very different results than those for a covariate on the location parameter only (not shown). Consequently and for reasons of parsimony, we examine only conditional extreme value distributions with a time-varying location parameter".
The model with constant coefficient of variation (CV) is an interesting alternative to the model that we present in our manuscript. However, investigating additionally this model would lead to a different and considerably extended manuscript. We feel that such a change is beyond the scope of this paper. The possibility of this model is discussed in the discussion and conclusions section (lines 399-401):
"A constant coefficient of variation as in Serago and Vogel (2018) would also be possible and as parsimonious as our model. In this case, a varying scale parameter linked to the location parameter would need to be implemented".

**General comment 4**

- One issue I would also suggest to discuss is the uncertainty in the covariates. The model used here assumes that the covariates are exactly known. If the uncertainty in their knowledge would also be included, would the flood quantile estimates still be more precise than for the classical GEV model?

**Response:** Thank you for this comment. In our manuscript we investigate only contemporaneous relationships between climate indices and flood peaks and do not focus on prediction. Our goal is mainly to identify spatial patterns of these relationships. For this reason we assume that covariates are exactly known. Of course if one wants to use the current model in a predictive mode, the uncertainty in the covariates must be additionally considered. A comment on this issue was added in the discussion (lines 413-415):
"The contemporaneous streamflow-covariate setup presented here can be used, together with a seasonal prediction of indices, for an ahead-season forecast of streamflow quantiles. In this case covariate uncertainty must be additionally considered".

Additional detailed comments:
Line 20: I would expect that the improvement of estimation of flood probabilities is conditional on how well the covariates can be predicted.

**Response:** The reviewer is right. This sentence was omitted in the revised manuscript.

Line 71: I am a little confused by the positive-negative anomalies vs. Northernsouthern Europe because the sentence terminates with "during its positive state".
Maybe a rephrasing could help.

**Response:** In the new version a change was made from "Particularly NAO has been shown to significantly influence the European winter climate with positive (negative) anomalies of moisture fluxes, cyclone passages and precipitation over northern (southern) Europe during its positive state" to "Particularly NAO has been shown to significantly influence the European winter climate: its positive state has been linked to positive (negative) anomalies of moisture fluxes, cyclone passages and precipitation over northern (southern) Europe".

Line 107: The motivation of using Bayesian inference because of the quantification of uncertainty sounds a bit weak. The quantification of uncertainties is possible also with other methods than Bayesian, which is instead usually selected when subjective preferences or prior information is available (at least by us... statisticians have more profound reasons).

**Response:** The reviewer is right. The reasons for choosing a Bayesian framework were better highlighted. The following sentences were added (lines 95-97):
"A Bayesian framework is adopted for the flood frequency analysis because of its advantages concerning the quantification and interpretation of uncertainty. Furthermore, prior information about hydrologic extremes exists in the literature and can be used for inference".

Line 150: The climate covariates are assumed exactly known in the method. Would it be possible to account for the fact that they are stochastic variables as well? I do not ask to change the method but maybe some discussion could be dedicated to this issue (see main comments).

**Response:** This point was answered in our reply to general comment 4.

Line 158: The motivation for assuming that only the location parameter vary in time, i.e., the brevity of records, is not very convincing. The Authors should discuss it more (see main comments).

**Response:** This point was answered in our reply to general comment 3.

Line 174: Since Bayesian inference is done here, there is no reason why priors should not be used. For the GEV shape parameter of the stationary model I recommend to use (at least) the "geophysical" prior in Martins and Stedinger (2000) (see main comments).

**Response:** This point was answered in our reply to general comment 2.

Line 174: Which non-informative priors are used? Not all of them would result in the same inference. For repeatability, they should be stated.

**Response:** We added a description of the prior distributions used: uniform priors for the location and scale parameters and a normal informative prior for the shape parameter (see also our reply to general comment 2.

Line 184: I would also look at the posterior distribution of the slope parameter and do the same as the Authors do here. I would just add a sentence to state that this is not a significance test (which has no meaning in Bayesian statistics).

**Response:** The comment was adopted. The following sentence was added (line 186): "Conditional models are considered as significant if the zero value is not included in the 90% posterior interval of the slope parameter (and thus not by means of a significance test)".

Line 216: I worry that, for engineering purposes, the estimates of 100-yr floods through GEV without accounting for regional information is not to be recommended.

**Response:** This point was answered in our reply to general comment 1.

Line 221: Starting the sentence with "Since a Bayesian framework is used" is confusing because it sounds like saying that uncertainties cannot be quantified with other methods too.

**Response:** The sentence was omitted in the revised manuscript.

Line 224 (and elsewhere): I would use the wording "posterior mode" instead of "maximum likelihood" because they may not be the same (it depends on the type of noninformative priors that are used). Bayesian posterior predictive distribution of flood peak quantiles or their posterior mean could have also been used. Is there a reason for choosing the posterior mode?

**Response:** The posterior mode was initially used in order to make results comparable with those of frequentist approaches. In the revised manuscript, the posterior median of flood peak quantiles was instead used because it is more representative of the posterior distribution.

Line 258: Is there any (even speculative) reason for the contradicting patterns for Scandinavia?

**Response:** In the revised manuscript a possible reason for these contradicting patterns is shortly discussed (lines 254-256).
"Scandinavian rivers usually have small catchments and are particularly fed by snowmelt in spring, subsequently in this area, both temperature and precipitation are important for runoff generation".

Line 265: I also believe that the coherence in space is indicative of a real signal, however spatial correlation of the flood time-series could be a nuisance here, meaning that one sees

the same dynamics in many sites because the same floods are occurring there (and therefore they should count as one site only). Since the spatial patterns are here over very large regions, the spatial correlation of the flood time-series cannot alone be responsible for it. However, I would suggest mentioning the problem.

**Response:** We agree that spatial correlation of floods plays a role for the detected coherence particularly for smaller regions, i.e. nearby gauges. We added a comment on the spatial correlation of the flood time-series (lines 261-264).
"It can be argued that these two latter cases could occur solely by chance or due to spatial correlation of nearby flood time series, however, results are coherent in space and cover large regions, which suggests a real influence of the circulation modes on the location parameter of the extreme value distributions, restricted though to certain sub-regions of Europe".

Line 299: One curiosity. Since the proposed model has constant variance (and the dependence of small and large floods on the covariate is the same in terms of the difference in m3/s) I suspect the relative difference to be affected by catchment area (meaning by the average flow in the river). Is it the case? Of course, since the model is fitted independently to every site, the differences in fitted shape parameters will make this relationship noisier.

**Response:** Since the difference between streamflow quantiles for high and medium covariate is normalized by the streamflow quantile for medium covariate we were not expecting that the catchment size plays a role in the percent relative differences. We assume that the high relative differences are due to a stronger influence of the climatic indices.

Section 3.2: I wonder how much the relative differences calculated here are due to the slope of the regression for the location parameter vs. the estimated shape parameters. Since no priors are used, I suspect that the posterior distributions of the shape parameter can be wide (spanning unreasonable values) and widely different between sites. Maybe a figure/table that also informs the reader about the obtained shape parameters would be useful.

**Response:** We added Table 4 with summary statistics of the shape parameter.

Figure 7: Shouldn't be the classical GEV the same within each column? The credible bounds look different. Have I missed something?

**Response:** We are sorry, this was a typo error noticed by the reviewer and was corrected.

Lines 335: The asymmetry of the credible bounds around the posterior mode is very well expected. If the posterior predictive distribution (or posterior mean) would have been used, that would have lied much more in the center of the bounds.

**Response:** The posterior median is used now. Indeed credible bounds are less asymmetric.

Line 352: I would add here a brief discussion on the predictability of the covariates since that is needed to make use of the model for prediction.

**Response:** This point was answered in our reply to general comment 4.

Line 357: I don't get the meaning of "...leads to highly varying flood quantile estimations for different probabilities of exceedance". Is the sentence referring to variations in time? Or space? Or between models with different covariates? And, finally, the variability for "different probabilities of exceedance" of flood peaks at one site exists in terms of relative differences. In term of difference in m3/s, there is no variability at all, since in the models only the location parameter can vary. Maybe I just misunderstood. A rephrasing could help.

**Response:** The variability concerns flood quantiles for the same station and probability of exceedance and for different values of the climate indices. It was rephrased to:
"For models with significant slopes, variations of the climate indices lead to highly varying flood quantile estimations for the same probability of exceedance".

Line 358: Related to my previous comment on line 299: is it because the catchments in North-West Scandinavia and Britain are smaller than the others? Or is it because of unreasonably large shape parameters of the GEV?

**Response:** The highly varying results in this area are in our opinion the result of a more important influence of the circulation indices. No influence of the catchment size was found.

Line 363: It is for me hard to see the decadal-scale variability in Figure 7. Maybe that could be shown in the figure.

**Response:** In the revised manuscript Fig. S5 was added in the supplementary material showing the evolution in time of the climate indices and their decadal-scale variability. We think that this will help the readers better interpret Fig. 7.

Line 402: As an additional challenge (on top of the three that the Authors have listed) I would add the fact that now covariates are assumed perfectly known and should be instead treated as stochastic variables, I think.

**Response:** This point was answered in our reply to general comment 4.

**Reply to Referee #2 Elena Volpi:**

The manuscript investigates the effectiveness of performing climate-informed extreme value analysis for flood probability estimation at the European scale. More specifically, the Authors analyze the effects of large-scale circulation patterns on seasonal extreme distributions by accounting for the relationship between extreme probabilities and climatic indices. As stated by the Authors, climatic indices are considered in recent literature works to justify or explain a non-stationary behavior depicted by extreme events.
In this regard, the innovative contribution of this paper is to perform a large-scale analysis, at a spatial scale that is "comparable" to that of the climatic indices considered in the work aiming at defining the conditional probability distribution of extreme flood events and proving coherent spatial patterns.
The manuscript is well written and organized; the methodology is almost well described, even if additional details could be included to help for reader understanding, and conclusions are well supported by results. Finally, within the Conclusion Section a detailed list of the limitations of the study is provided. Summarizing, the topic is of interest for the scientific community and the manuscript deserves to be considered for publication in this Journal. I have some comments about the work that are listed in the following paragraph; I hope that they will be helpful for manuscript improvement.

**Response:** We would like to thank Elena Volpi for her comments. In the revised manuscript we followed most of the reviewer's recommendations, since this definitely improved our study. Below, we provide justification for some suggestions that we did not follow. We saw from the comments of the reviewer that some parts of our study needed a more detailed explanation. In our revised manuscript we provide these additional details.

**General comment 1**
1. The Authors hint in the Introduction Section at the nonstationary framework incorporating climatic indices into flood frequency analysis, but they do not make a clear distinction between periodicity (or cyclo-stationarity) and trends (in the mean or variance). For the sake of clarity, this could be discussed from the very beginning of the manuscript (e.g. at line 49). Are the Authors assuming stationarity which is a "prerequisite to make inference from data", as discussed in detail by the cited papers by Koutsoyiannis and Montanari (2015) and Serinaldi and Kilsby (2015)?
**General comment 2**
2. At line 136 the Authors define the model driven by climatic indices as "climatic informed model", justifying this choice based on the fact that "if covariates have a stochastic structure and no deterministic component, the resulting distribution is not truly nonstationary". I do agree on this, as the Authors states at line 135 that the climatic indices are stochastic process not showing clear trends. But, it is expected they are characterized by persistence and/or periodicity. A detailed description of the stochastic behavior of the climatic indices is missing in the manuscript, while they are clearly described from a physical point of view (lines 59-90). E.g. which are the relevant time-period and is the period covered by observations long enough to catch climatic indices periodicities?

**Response to general comments 1 and 2:**
We want to thank the reviewer for these interesting comments. In our manuscript we acknowledge the issue that models conditional on time-varying covariates with a stochastic structure can be stationary, even if the probability density function changes in consequent years. However, we feel that addressing the issue of stationarity/nonstationarity (and thus ergodicity) and the stochastic structure of the covariates in adequate detail would change considerably the focus of our manuscript and we prefer not to make this addition. For this reason we chose the term "classical" and "climate-informed models" and we do not refer to stationary/nonstationary models. We will consider going more in this direction in our future work.

**General comment 3**

3. Even if the aim of the work is to find results at the European scale, I would suggest the Authors to add a figure showing results for a single station, as an illustrative example to explain the methodology and the rationale behind it (e.g. the structure of the climatic informed GEV). Similar to figure 7, it could be of interest to show the evolution in time of the climatic indices (see comments 2) and the performance of classical GEV and climatic-informed GEV, especially for quantile extrapolation, with uncertainty bounds.

**Response:** We adopted the recommendation of the reviewer and, additionally to the results for three specific stations shown in Fig. 7, we show in Fig. 8 the performance (point estimates and uncertainty bounds) of the classical GEV and the climatic-informed GEV when plotted against the covariate values (here the extrapolation towards more extreme index values and not lower probabilities of exceedance is shown). We feel that this further clarifies our methodology. We did not opt to include an additional figure explaining the climate-informed model because, as we also state in the introduction, during the last years there have been many studies applying such a conditional framework to single or a few stations. We feel there is enough published material explaining this methodology. The suggestion to show the evolution in time of the climate indices was adopted and these are illustrated in Fig. S5 of the supplementary material.

4. If I understand correctly, conditional models preferred o classical GEV in Table 1 are those respecting both criteria (minimum value of DIC and significantly different from zero coefficient of linear variation with the climatic indices); this could be highlighted in the result section from the beginning of the section. The number of times (stations) each conditional model is preferred with respect to classical GEV is not so high, being in the best case the 44% and on average at about 20%. The use of two criteria does not seem to affect this result much (as in lines 276-280); hence, the evidence of the climatic informed model does not appear to be very strong, even if clear spatial patterns emerge. The latter is the more relevant result, based on my opinion, and this should be stressed in the abstract and conclusion sections.

**Response:** We adopted this suggestion and in the revised manuscript we highlighted the selection criteria in the results section (line 246). Furthermore, in the discussion and conclusions section we highlighted that the effect of each index independently affects on average a 20% of the database (line 351). However, the number of stations affected by at least one index significantly is much higher, especially in winter. We feel that this is a result that indicates a real influence of the circulation indices to the streamflow extremes.

5. Since spatial patterns are influenced by correlations among climatic indices (that are illustrated in the supplementary material as spatial maps), I suggest the Authors to report in the manuscript a table summarizing cross-correlations among the indices (even if they are not an exhaustive measure of the underling complex physical phenomena).

**Response:** We adopted this suggestion and we added Table S1 in the Supplementary material summarising linear correlations between the seasonal indices.

6. Lines 276-279. DIC is a measure of model evidence; even if the climatic informed model has a smaller value of DIC with respect to classical GEV, the difference among the two values is probably not enough to results in a "strong evidence" of the first model compared to the second one. See, e.g., Kass and Raftery (1995) where two different interpretations of the Bayes factor are provided.
• Kass, R. E., Raftery, A. E. (1995). Bayes factors. Journal of the american statistical association, 90(430), 773-795.

**Response:** Indeed when information criteria are used for model comparison, the difference of two values does not always provide "strong evidence" for model choice. Here since we are

using two criteria, the DIC and slope significance, we feel that the evidence for model selection.

7. Figure 7 compares conditional (climate informed) and unconditional quantiles considering p=0.01 for three stations. It should be clearly stated that conditional quantiles are computed in this case based on the observed values of the climatic indices year by year.

**Response:** We adopted this suggestion. A sentence stating this year by year quantile calculation was added (Lines 319-320): "Conditional quantiles are calculated on a year-to-year basis, based on the observed values of the selected climate indices".

8. As the climate informed models have a larger number of parameters (one more in this case) to be estimated based on data, it is expected that their uncertainty bounds are larger than those provided by classical GEV. In other words, nonstationarity flood frequency analysis adds an additional component of uncertainty if the model between parameters and covariates is estimated from data and not fully a-priori defined based on additional physical information (Serinaldi and Kilsby, 2015). However, this is not what emerges from figure 7. This issue should be clarified.

**Response:** This was a very helpful comment. We realised that Fig. 7 and the conditional/unconditional uncertainty bounds needed to be further discussed. Fig. 7 does not contradict the findings and discussion of Serinaldi and Kilsby (2015). The range of the uncertainty bounds is an interplay between the model complexity and the additional information provided by the more complex models. In general, more complex models not providing extra information are expected to lead to an increase in uncertainty. More complex models providing "adequate" additional information are expected to lead to decreased uncertainty.
In order to explain this relation better, we added Fig. 8 comparing streamflow quantiles for the classical and a conditional model for a single station. Point estimates and credibility intervals of streamflow quantiles for a probability of exceedance 0.02 are plotted versus NAO values. The following paragraph was added (Lines 334-340):
"The range of uncertainty bounds reflects an interplay between model complexity and the additional information provided by the more complex models. In Fig. 7, uncertainty bounds are narrower in the case of the "best" conditional models (e.g. subplot A1). Uncertainty increases when extrapolations are made towards high and low index values. This can be more easily observed in Fig. 8. For the classical case, the range is about 94 $m^3$/s. For the climate-informed case and NAO = 0 (close to its median value) the range is around 70 $m^3$/s. The range increases to 74 $m^3$/s for NAO = 1 and to 80 $m^3$/s for the most extreme observed NAO value (NAO = -2.1). For a NAO value around 3/-3 the range of uncertainty bounds reaches that of the classical model".

9. Lines 329-330. This should be true if the climate indices can be accurately predicted. The issue should be discussed further since it is closely related to the implications of the results presented in the paper for practical applications. Furthermore, I'm asking myself if the improvement in flood quantile estimate at the local scale thanks to climate indices is really significant from a practical point of view given the large uncertainty that characterizes all the estimates (fig. 7); I would like to read a comment on this from the Authors.

**Response:** A comment was added in the discussion explaining the effect of the index uncertainty in the limitations of the study. Indeed if one wants to predict the indices in order to use them for the estimation of streamflow quantiles, uncertainties will be higher. In lines 413-415 we state: "The contemporaneous streamflow-covariate setup presented here can be used, together with a seasonal prediction of indices, for an ahead-season forecast of streamflow quantiles. In this case covariate uncertainty must be additionally considered".
Concerning the uncertainty of flood quantiles

I our opinion the use of the climate-informed model is significant from a practical point of view, especially when point estimates of flood quantiles and their uncertainty bounds strongly diverge between the classical and climate-informed models (as e.g. in Fig. 7, 8). However, for low exceedance probabilities, uncertainties are large and this can be improved with the use of a regional framework. We discuss this issue in lines 384-389 of the discussion and conclusions section: "In this study, a local, site-specific flood frequency model was developed. This model allowed to identify spatial coherence in relations between streamflow extremes and large-scale atmospheric patterns. However, a shortcoming of this methodology is the high uncertainty of streamflow estimates for high probabilities of exceedance (corresponding for example to the 100- or 200-year flood). Instead of a local framework, a regional framework can be alternatively implemented. The latter, by considering all available streamflow information in a region, decreases uncertainty and offers the possibility of improving streamflow quantile estimation".

10. Line 58. The Authors could also consider the recent paper from Serinaldi et al. (2018) discussing limitations of nonstationary detection based on trend tests.
• Serinaldi, F., Kilsby, C. G., Lombardo, F. (2018). Untenable nonstationarity: An assessment of the fitness for purpose of trend tests in hydrology.

**Response:** Thanks for this very interesting paper. We included it in the discussion of limitations of conditional / nonstationary models.

11. Line 71. A reference is needed.

**Response:** It was added.

12. Please definite t after eq. (3).

**Response:** It was defined (Line 130).

13. Line 173. Please define what is meant by non-informative priors in this case. If the non-informative prior is a uniform distribution, its support (range of variability of the random parameter) could have effects of posterior distribution and evidence estimation.

**Response:** A description of the prior distributions was added. In the revised manuscript uniform priors were used for the location and scale parameters and a normal informative prior for the shape parameter.

14. Eq. (8). y or Y ?

**Response:** We replaced y with Y.

15. Line 194. Please define $\bar{\theta}$.

**Response:** The definition was added.

16. Line 219. Are the Authors assuming a Gaussian (marginal) distribution for climatic indices? The assumptions on those variables and their stochastic behaviour are not clear (see also general comments 2 and 3).

**Response:** We are currently not making an assumption about the marginal distribution of climate indices. Figure S6 was added in the supplementary material showing the histograms of all seasonal indices.

**Reply to Referee #3 Francesco Marra:**

This study presents a methodology to assess and quantify the impact of climatic covariates in the estimation of time-dependent flood probabilities. The method is tested on a wide sample of catchments in Europe. The paper is clearly and concisely written and the topic is of interest for the readers of this journal.
In my opinion, the study should be seen as an additional step in the efforts of the hydrological community towards a better understanding and quantification of flood risk and, as well underlined by the authors, the spatial consistency of the results indicates some degree of significance in the adopted model. However, further steps are required before the suggested method can be effectively applied in practice. Both the reviewers before me pointed out very interesting comments, many of which I happened to share. I come last so I'll try not to overlap.

**Response:** We would like to thank Francesco Marra for his comments. The referee has shared very valid concerns that we hope to address in the revised version of our manuscript.

**General comment 1**
In general, my main concern derives from the GEV approach that requires a number of hypotheses and, if not integrated within a regionalization framework, is prone to extremely large uncertainties.

**Response:** The concern of the reviewer is valid. Indeed, a regional framework is commonly used in order to extrapolate inference to higher return periods. Here, focusing more on identifying coherent patterns in space, we used a local framework, which is able to recognise significant influence of certain indices to the extreme streamflow quantiles in certain regions of Europe. In order to reduce uncertainty, we constrained our analysis to the 50-year return period which covers the data length: in our study the overlapping period between climate indices and streamflow time series is between 50 and 70 years. Furthermore, in order to improve GEV fits, prior information for the shape parameter was included in the analysis (see also our response to main comment 2 of Alberto Viglione).

In addition to the references recommended by Elena Volpi, I may suggest the reading of Marani and Ignaccolo (2015), that provide a different perspective on extreme value analysis and the GEV approach (potentially for nonstationary extremes) that deserves attention.

**Response:** Thanks for this very interesting paper. We will consider it for our future work.

To conclude, I think the paper definitely deserves publication, but some more discussion and comments on the adopted methods are required. Potentially, some additional analyses could be of help. Below my detailed comments.

• Is the use of climate-informed models contradicting the identical-distribution assumption behind the use of GEV? This perhaps needs to be discussed.

**Response:** The condition of independent and identically distributed observations of GEV can be relaxed to include parameters conditioned on time-varying covariates. This can be achieved by converting the original data to generalized "residuals" that are identically distributed (Katz et al., 2002). A comment on this issue was added in the revised manuscript (Lines 123-124): "For the climate-informed models the condition of independent and identically distributed observations of the classical GEV is relaxed to include parameters conditioned on time-varying covariates (Katz et al., 2002)".

• The inclusion of climate information in the model raises the number of parameters to be estimated to 4. Is there a risk of overfitting?

**Response:** Our climate-informed models have one more parameter compared to the classical GEV, derived by physical reasoning (the climate indices influence the climate and hydrology of the European area). According to e.g. Katz et al. (2002), such models are reasonable. Furthermore, our conditional models are compared to the classical GEV by means of the DIC, which penalizes model complexity. We are thus confident that the possibility of overfitting is minimized.

• How the authors explain that the linear model applied to the scale parameters (rather than location) provides similar results? Shouldn't the two parameters be related one another since the location is related the mean and the scale to the variance of the annual maxima? Is it correct to change one of them and keep the other fixed?

**Response:** The reviewer is right, indeed both the location and scale parameter are expected to change based on the climate state. However, we tested for significance of varying scale parameter by running the model with both location and scale variable. This preliminary study showed only few cases with significant slopes of the scale parameter. For this reason and for reasons of parsimony, we decided to keep the scale parameter stable and to condition only the location parameter on the climate indices (see Lines 149-152). In the discussion and conclusions sections we further comment on the possibility of a model with a constant coefficient of variation as an alternative to our model (Lines 398-401):
"Furthermore, we also assumed a varying location parameter and constant scale parameter. A constant coefficient of variation as in Serago and Vogel (2018) would also be possible and as parsimonious as our model. In this case, a varying scale parameter linked to the location parameter would need to be implemented".

• The GEV approach is highly sensitive to the shape parameter, which is prone to large estimation uncertainty when derived from short data records (50 years), particularly when using the maximum likelihood method. Why not using an Lmoments estimation method? Could the inclusion of prior information on the GEV shape parameter improve the accuracy of the results? Perhaps this aspect should be addressed in the study to check consistency in the significant indices (in the end shape and scale are then used as prior information in the estimation of the climate-informed model parameters).

**Response:** Thanks for this interesting comment. The Bayesian approach was chosen because of its advantages concerning quantification of uncertainty. A second reason is the possibility to include prior information for model fitting (see Lines 95-97). The choice of a likelihood-based method offers additionally a straightforward way of including covariates in the frequency analysis. Indeed inclusion of prior information on the GEV shape parameter improves the accuracy of the results. In our revised manuscript we used an informative prior for our classical and climate-informed GEV. Please see our reply to general comment 2 of Alberto Viglione for more details on the prior distribution chosen. The posterior distributions of the parameters of the classical GEV are not currently used as priors for the climate-informed case. Each model is fitted independently. A comment on this issue was added in Lines 168-169.

• A linear model to relate climatic indices and GEV location parameter is chosen. Clearly, more complex models are not recommended due to the limited data sample and overfitting problems, but this represents a simplification of reality. How can this affect the results? This should be discussed.

**Response:** In our revised manuscript we extended our discussion on alternative models to point out more clearly the limitations because of choosing a linear model (Lines 391-398):
"A linear relationship was assumed between streamflow extremes and the large-scale atmospheric indices. This is a simplification of reality and some relations may be over- or underestimated due to existing non-linearities in the climate-streamflow system. More complex, particularly non-linear models would also be possible candidates for describing the

relation between climate indexes and flood probabilities. However, with increasing model complexity, the chances for model overfitting also increase. In this study we assumed a symmetric influence of the positive and negative phases of the climate indices. However, an asymmetric relation may better describe the effect of certain climate modes on streamflow extremes. For example, Sun et al. (2014) used an asymmetric piecewise-linear regression to account for the different effects of El Niño and La Niña on rainfall extremes in Southeast Queensland, Australia".

• What do the authors recommend for situations in which more than one climatic index is significant?

**Response:** A multi-linear model for such cases is also possible. For example, in our study there seems to be a significant influence of both NAO and SCA in winter in Central Europe. We comment on this issue in our discussion (Lines 403-408):
"Single covariate models were developed, focusing on the separate effect of each individual climate mode. The methodology can be extended to a model considering several covariates at the same time. In that case, dependencies between the covariates, if existent, should be taken into consideration. López and Francés (2013) overcame this problem by using the principal components of climatic indices as covariates for the flood frequency analysis. This, however, increases the model complexity considerably and thus the chances of model overfitting. This needs to be considered in developing models with multiple covariates".

Minor comments:
- Lines 24-28 in the abstract are not easy to read, I suggest to rephrase them;

**Response:** We rephrased these sentences. In the revised manuscript we state: "For certain regions, such as Northwest Scandinavia and the British Isles, yearly variations of the mean seasonal climate indices result in considerably different extreme value distributions and thus in highly different flood estimates for individual years that can also persist for longer time periods".

- Introduction: the proposed method is of interest for (re-)insurance applications and for flood risk management. I think the design applications are not interested since year-by-year variability is not relevant

**Response:** Indeed the proposed framework makes more sense for (re-)insurance purposes and flood risk management. We omitted "engineering design" in line 44.

- 169-172: please provide more details for readers not familiar with the technique;

**Response:** More details on the No-U-Turn Sampler-Hamiltonian Monte Carlo approach. In lines 163-165 we state: "NUTS is an extension to HMC, a Markov chain Monte Carlo (MCMC) algorithm that avoids the random walk behavior and sensitivity to correlated parameters which characterize many MCMC methods (Hoffman and Gelman, 2014)".